# Intensive grassland management disrupts below-ground multi-trophic resource transfer in response to drought

Mathilde Chomel [1,13] ✉, Jocelyn M. Lavallee [1,2,3], Nil Alvarez-Segura [4,5], Elizabeth M. Baggs [6], Tancredi Caruso [7,8], Francisco de Castro [7,9], Mark C. Emmerson[7], Matthew Magilton [7,10], Jennifer M. Rhymes [1,11], Franciska T. de Vries [1,12], David Johnson [1] & Richard D. Bardgett [1]

Modification of soil food webs by land management may alter the response of ecosystem processes to climate extremes, but empirical support is limited and the mechanisms involved remain unclear. Here we quantify how grassland management modifies the transfer of recent photosynthates and soil nitrogen through plants and soil food webs during a post-drought period in a controlled field experiment, using in situ $^{13}C$ and $^{15}N$ pulse-labelling in intensively and extensively managed fields. We show that intensive management decrease plant carbon (C) capture and its transfer through components of food webs and soil respiration compared to extensive management. We observe a legacy effect of drought on C transfer pathways mainly in intensively managed grasslands, by increasing plant C assimilation and $^{13}C$ released as soil $CO_2$ efflux but decreasing its transfer to roots, bacteria and Collembola. Our work provides insight into the interactive effects of grassland management and drought on C transfer pathways, and highlights that capture and rapid transfer of photosynthates through multi-trophic networks are key for maintaining grassland resistance to drought.

Grasslands are under threat from ongoing degradation caused by multiple co-occurring drivers, including management intensification and climate extremes[1]. These climate extremes, such as drought events, are a recurring phenomenon in many ecosystems and are predicted to increase in frequency and intensity in the coming decades[2,3]. A major challenge therefore is to understand the interactions between management intensification and drought to inform

sustainability policy aimed at protecting the multiple ecosystem services that grasslands provide[1,4]. Many studies have demonstrated that intensive grassland management, characterised by the regular use of inorganic fertilisers and high livestock stocking densities, can decrease plant diversity[5], the abundance and diversity of larger body-sized earthworms, nematodes and microarthropods[6,7], and the biomass of saprotrophic and arbuscular mycorrhizal (AM) fungi[8]. Such changes

[1]Department of Earth and Environmental Sciences, The University of Manchester, Manchester M13 9PT, UK. [2]Department of Soil and Crop Sciences, Colorado State University, Fort Collins, CO, USA. [3]Environmental Defense Fund, 257 Park Ave S, New York, NY, USA. [4]Institute of Biological and Environmental Sciences, University of Aberdeen, Cruickshank Building, Aberdeen AB24 3UU, UK. [5]Department of Climate Change, EURECAT, Technological Centre of Catalonia, Amposta, Spain. [6]Global Academy of Agriculture and Food Systems, Royal (Dick) School of Veterinary Studies,  University of Edinburgh, Midlothian, UK. [7]School of Biological Sciences and Institute for Global Food Security, Queen's University of Belfast, Belfast, UK. [8]School of Biology and Environmental Science, University College Dublin, Dublin, Ireland. [9]AgriFood & Biosciences Institute, Belfast, UK. [10]School of Life Sciences,  University of Lincoln, Lincoln, UK. [11]UK Centre for Ecology and Hydrology, Environment Centre Wales, Deiniol Road, Bangor LL57 2UW, UK. [12]Institute for Biodiversity and Ecosystem Dynamics, University of Amsterdam, Amsterdam, the Netherlands. [13]Present address: FiBL France, Research Institute of Organic Agriculture, 26400 Eurre, France. ✉ e-mail: mathilde.chomel@fibl.org

also have important consequences for biogeochemical cycles given that management-induced shifts in soil food web properties, including changes in the relative abundance of bacteria and fungi[6,7,9], can predict processes of carbon (C) and nitrogen (N) cycling[10–12]. Specifically, shifts to bacterial dominated food webs resulting from management intensification have been linked to faster rates of nutrient mineralisation and greater losses of C and N from soil following perturbations such as drought[10, 13–15]. For this reason, the fungal/bacterial biomass ratio is commonly used as an indicator for the relative activity of the two groups in channelling energy and matter through soil food webs, albeit focussing on bacteria and fungi rather than the whole soil food web[14]. It is becoming apparent, however, that the stability of ecosystem functions can only be understood if multiple trophic levels and interactions among them are considered[16,17]. Moreover, there is growing awareness of the importance of rhizodeposition as a driver of below-ground energy flow and the structure and functions of soil food webs[18–23], although our understanding of these biotic interactions and their functional significance remains limited. Consequently, there is a need for improved understanding of the role of food web structure from a multitrophic perspective, including plants and below-ground organisms, in modulating the response of processes of C and N cycling to perturbations, such as climate extremes.

Evidence is mounting that changes in food web structure play an important role in regulating the stability of soil functions following perturbations and could impair the ability of soil food webs to resist and recover from future extreme climatic events. Theoretical studies, for example, predict that a modification of soil food web architecture based on energy channels can modify the stability of a food web and its ability to resist large perturbations[24–27]. Recent empirical studies also indicate that intensive management can decrease the resistance to drought of plant productivity and soil respiration[28, 29], soil food web biomass[28], and C allocation to soil microbial communities[30]. Despite this knowledge, our understanding of the key components and interactions within food webs that drive the response of C and N fluxes to perturbations is limited. Furthermore, while we have a good understanding of the response of ecosystem processes during drought, comparatively little is known about ecosystem responses, including C and N flux, after these events during a post-drought period[31]. These gaps in our knowledge hamper our ability to make reliable predictions of soil C and N cycling and sequestration in response to dual pressures of agricultural intensification and increased frequency and intensity of climate extremes.

Here, we experimentally investigated how grassland management modifies the transfer of recent photosynthates and soil N through plants and the soil food web during a post-drought period. Because intensive management modifies plant and soil communities, we expected intensive management to disrupt the coupling of C flow from plants to mycorrhizal fungi and the soil food web leading to: a) greater soil C and N losses as $CO_2$ and $N_2O$, because we expected shifts to bacterial-dominated food webs that are linked to faster rates of nutrient mineralisation; and b) a greater legacy effect of drought on C and N fluxes because we expected that intensive management decreases the stability of the soil food web and its ability to resist to drought.

We tested these hypotheses by simulating summer drought using rainfall shelters on paired extensively and intensively managed mesotrophic grasslands across three geographically distinct locations in northern England with similar soils, vegetation and topographical conditions (Supplementary Figs. 1 and 2). In both control and drought plots, plant communities and soil food web (microorganisms and mesofauna) biomass and composition were determined. To simultaneously investigate the short-term allocation of N and recently assimilated C below-ground during a post-drought period, we pulse-labelled plants with $^{13}C$-$CO_2$, and soil with $^{15}N$-$NO_3$ at the end of the simulated drought. We traced the incorporation of $^{13}C$ and $^{15}N$ into plant shoot and roots, microbial biomarkers, soil mesofauna trophic groups and soil $CO_2$ and $N_2O$ fluxes over a 20-day post-drought period (Fig. 1).

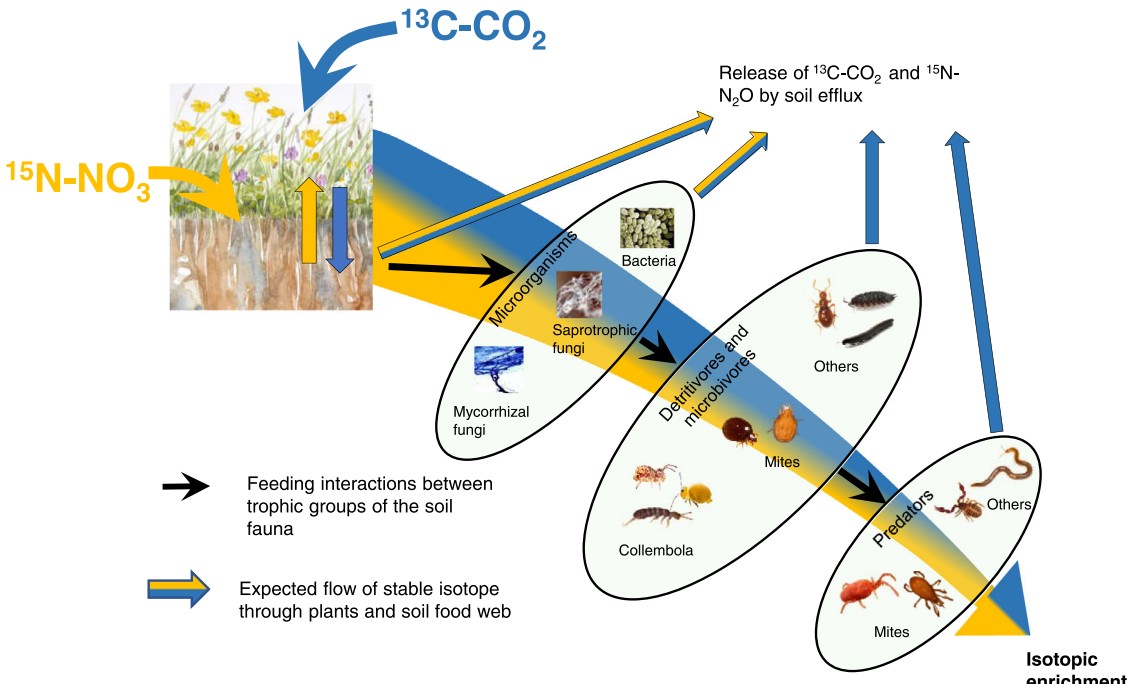

**Fig. 1 | The expected paths of C and N tracers in the plant-soil system.** Conceptual diagram of the expected flow of $^{13}C$ (blue) and $^{15}N$ (yellow) through plants and soil trophic groups. The figure represents the groups we analysed for their $^{13}C$ and $^{15}N$ content, hence the simplification of the food web. The ellipses represent the trophic groups, and although interactions within trophic groups can happen, the main flow of stable isotope will be from one trophic group to another (represented by arrows). The isotopic enrichment is expected to decrease at higher trophic levels of the food web. Grassland illustration by J. C. Bardgett, pictures from P. Lebeaux (www.animailes.com) and M. Chomel.

Our results show significant drought legacy effects on C transfer below-ground in intensively managed grassland, as evidenced by an increase in plant shoot C uptake and a decrease of the transfer of recent photosynthate to roots, bacteria, and Collembola during the post drought period. In contrast, extensive grassland management provided greater potential to buffer the legacy effect of drought on the transfer of C below-ground, most likely by promoting the transfer of recent photosynthates to mycorrhizal fungi and fungivorous mites.

## Results

### Effect of grassland management on the response of C and N flow to drought

During rain exclusion, soil moisture at the one site where it was measured continuously was reduced on average by $56 \pm 0.4\%$ in intensively managed grassland and $74 \pm 0.4\%$ in extensively managed grassland during the last 27 days of the drought (Supplementary Fig. 3). At this site, the post-drought recovery of soil moisture was slower in the extensively compared to the intensively managed grassland. Sequential soil moisture measurements across all three sites showed no significant differences of the soil moisture reduction between managements at the end of the drought but confirmed the slower recovery of soil moisture in extensively managed compared to intensively managed grassland (Supplementary Fig. 4).

In extensively managed grassland, drought increased the biomass of detritivorous mites and predatory mites, but reduced the biomass of actinobacteria (as assessed by PLFA) and Collembola compared to the control (Supplementary Fig. 5, Supplementary Table 1). In intensively managed grassland, drought increased the biomass of microbes and detritivorous mites, but reduced the biomass of plant shoots and other detritivores in comparison to the control (Supplementary Fig. 5, Supplementary Table 1).

In extensively managed grassland, drought had no detectable legacy effect on the uptake of $^{13}C$ by plants, its transfer to roots and microorganisms, or soil $^{13}C$-$CO_2$ efflux (Fig. 2, Supplementary Table 1), but reduced the transfer of $^{13}C$ to Collembola only compared to the control. However, in intensively managed grassland, drought had a stronger impact and increased plant shoot $^{13}C$ enrichment and $^{13}C$ relative enrichment of soil $CO_2$ efflux, but decreased $^{13}C$ transfer to roots, bacteria and Collembola compared to the control (Fig. 2, Supplementary Table 1). Drought had no legacy effect on N fluxes to the plants and soil communities in both grassland managements, but only reduced $^{15}N$ transfer as $N_2O$ efflux in extensively managed grassland compared to the control (Fig. 2, Supplementary Table 1).

### Effects of grassland management on soil food web composition and soil functioning

Grassland management modified plant communities and soil properties, with greater above-ground plant biomass, soil pH, nitrate concentration, and bulk density, and lower water holding capacity and below-ground plant biomass in intensively managed compared to extensively managed grasslands (Supplementary Fig. 6). The influence of land management on soil $CO_2$ efflux was not consistent through the sampling period (interaction management*time, $F = 2.86$, $P = 0.0162$, Supplementary Fig. 7), with higher $CO_2$ efflux in extensively managed compared to intensively managed grassland at time 0 and the opposite at day 20 (Supplementary Fig. 7). There was a trend of higher soil $N_2O$ efflux in extensively managed grassland compared to intensively managed grassland on the first day after the pulse labelling, but this was not significant (interaction time*management, $P = 0.08$, Supplementary Fig. 7).

In terms of soil community composition, across all sites and treatments, Collembola was the most abundant group of mesofauna ($30,949 \pm 3,718$ individuals m$^{-2}$, 80% were within the entomobryomorpha group), followed by detritivorous mites ($25,114 \pm 2,772$

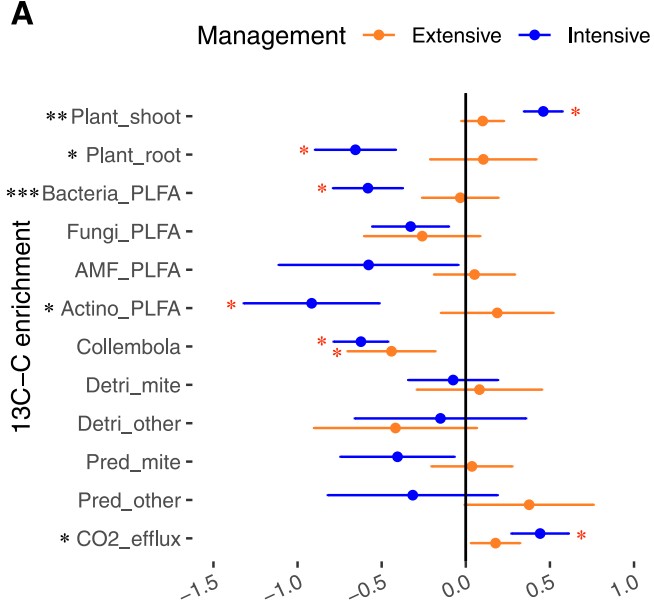

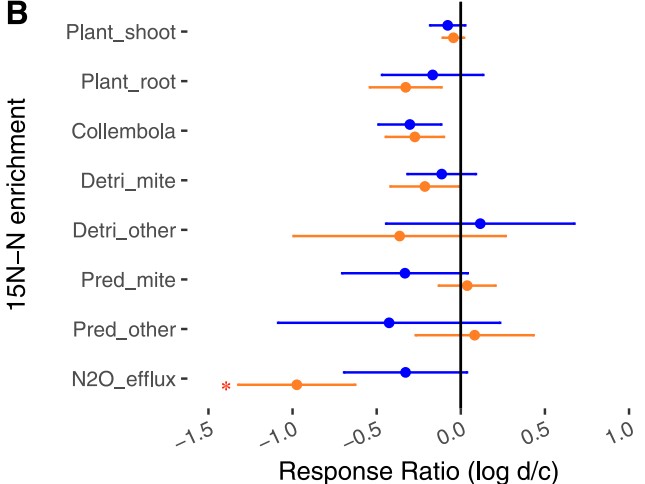

**Fig. 2 | Management impact on the drought effect on tracer transfer in the plant-soil system.** Response ratio of the post-drought effect on the $^{13}C$ (A) or $^{15}N$ (B) enrichment of the different pools as a function of grassland management (log (drought/control)). The sign (positive or negative) of the logRR corresponds to the direction of the drought effect on the $^{13}C$ or $^{15}N$ enrichment, while a response ratio of zero indicates no post-drought effect. This figure only highlights the strongest effects as logRR values have been calculated on averaged $^{13}C$ or $^{15}N$ enrichment across all sampling dates. Dots represent mean $\pm$ SEM ($n = 18$). Significance for management effect on the logRR from two sided linear mixed-models are reported with ***$P < 0.01$, **$P < 0.01$, *$P < 0.05$. Red asterisks indicate significance for drought effect by the examination of the confidence intervals of predicted means from the linear mixed-effects models, see Supplementary Table 1 for details. amf_PLFA AM fungal PLFA, actino_PLFA actinobacteria PLFA, detri_mites decomposer mites, detri_other other decomposers, pred_mite predatory mites, pred_other other predators.

individuals m$^{-2}$, 88% were oribatids) and predatory mites ($20,094 \pm 2,279$ individuals m$^{-2}$). Soil community composition differed significantly between extensively and intensively managed grasslands (Fig. 3, Supplementary Table 2, PERMANOVA, $F = 5.12$, $P < 0.001$). Despite this finding, there was substantial overlap in community composition (Fig. 3) and we detected significant differences in the distribution of data, with more multivariate dispersion in soil communities in extensively compared to intensively managed grassland

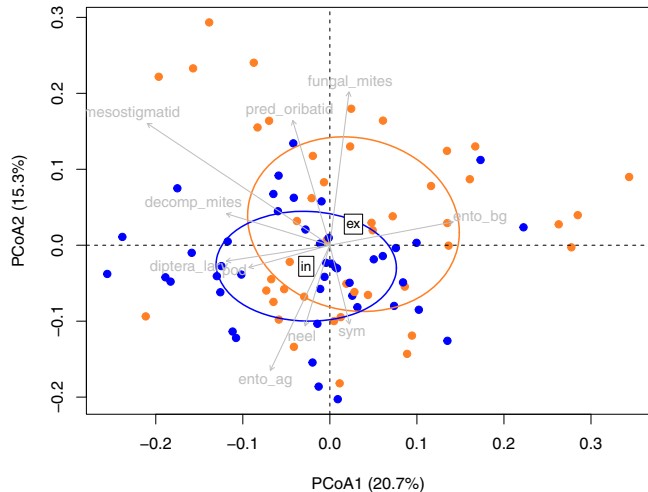

**Fig. 3 | Grassland management impact on soil food web.** Principal coordinates analysis (PCoA) of soil food web communities from grasslands with different land management (extensive in orange, intensive in blue) from the control plots at 5 sampling date ($n = 90$). The PCoA was based on Bray–Curtis distance using soil mesofauna abundances (individuals m$^{-2}$) and PLFA biomass (g m$^{-2}$). Grey arrows show the correlation vectors of the variables with the ordination. Significance was tested by Monte Carlo permutation against 999 random datasets and variables with $P > 0.05$ were kept (see Supplementary Table 2). Ento_ag epigeic entomobryomorpha, ento_bg eudaphic entomorbyomorpha, pod poduromorpha, sym symphypleona, neel neelipleona, decomp_mites decomposer mites, fungal_mites fungivorous mites, pred_oribatids predatory oribatids, pred_col_ad predatory coleoptera, pred_col_larv predatory coleoptera larvae, detrit_col_ad detritivorous coleoptera, detrit_col_larv detritivorous coleoptera larvae, diptera_larv diptera larvae, bactplfa bacterial PLFA, fungplfa fungal PLFA, amfplfa AM fungal PLFA, actinoplfa actinobacteria PLFA.

(PERMDISP, $F = 9.1$, $P = 0.033$). The biomass of detritivorous mites (mainly Oribatids) was smaller ($F = 7.03$, $P = 0.0096$), while the biomass of other decomposers (mainly diptera larvae) was greater ($F = 9.08$, $P = 0.0034$) in intensively managed compared to extensively managed grassland (Supplementary Fig. 8). Grassland management did not have a significant impact on bacterial, fungal, AM fungal and actinobacteria biomass, nor the fungal/bacterial ratio, across the three sites (Supplementary Fig. 8).

### Effects of land management on $^{13}$C and $^{15}$N flow
Across the three sites, plant shoots in the intensively managed grassland were significantly less enriched in $^{13}$C than in extensively managed grassland (Fig. 4, Supplementary Table 3, $P = 0.0015$). The maximum $^{13}$C-enrichment in plant shoots occurred at the end of the labelling period and was (on average) 0.55 and 0.39 atom % excess in extensively and intensively managed grassland, respectively. The enrichment decreased substantially 1 day after labelling to an average of 0.13 and 0.11 atom % excess (Fig. 4, Supplementary Table 3). The uptake of $^{15}$N in plant shoots was unaffected by grassland management, but increased gradually over two days after the labelling and reached a plateau of 4.5-5 atom % excess (Fig. 5, Supplementary Table 3). Land management had no detectable influence on the $^{13}$C- or $^{15}$N-enrichment of plant roots (Figs. 4 and 5, Supplementary Table 3).

Overall, among soil microorganisms, saprotrophic fungi had the greatest enrichment of $^{13}$C, whereas among soil fauna, Collembola had the greatest enrichment of $^{13}$C (Fig. 4). Intensive management decreased $^{13}$C enrichment in arbuscular mycorrhizal (AM) fungi and detritivorous mites compared to extensively managed grassland (Fig. 4, $P < 0.05$, see Supplementary Table 3), and no management effects on $^{15}$N enrichment of soil fauna were detected (Fig. 5, $P > 0.05$, see Supplementary Table 3). Although there was a greater

$^{13}$C-enrichment of the soil $CO_2$ efflux in extensively managed compared to intensively managed grassland ($P < 0.001$, Fig. 5, Supplementary Table 3), no difference in $^{15}$N-enrichment of $N_2O$ emissions was detected.

### $^{13}$C and $^{15}$N allocation
One day after labelling, 26% of $^{13}$C initially fixed by plant shoots remained in the shoots and 2.6% was recovered in the roots (Supplementary Fig. 9, Supplementary Table 4). A similar pattern was observed for the $^{15}$N tracer, for which 39% was on average recovered in the plant shoots and 12% in the plant roots (Supplementary Fig. 9, Supplementary Table 4) one day after the labelling. Intensive management decreased the $^{13}$C recovery in plant roots but increased the $^{15}$N recovery in plant shoots (Supplementary Fig 9, Supplementary Table 4).

Analysis of $^{13}$C and $^{15}$N pools revealed that in extensively managed grassland, soil fauna tended to store more photosynthates-derived C and fertiliser-derived N compared to intensively managed grassland (Supplementary Fig. 10, Supplementary Table 5). This pattern becomes even clearer when considering the relative allocation (Supplementary Fig. 9, panel C). Indeed, extensive management increased the $^{13}$C recovery in AM fungi, detritivorous mites and predatory mites, and the $^{15}$N recovery in detritivorous mites (Supplementary Fig. 9, Supplementary Table 4, $P < 0.05$). Drought only decreased the recovery of the $^{15}$N in the plant roots (Supplementary Fig. 9, Supplementary Table 4, $P < 0.001$) and had no significant effect on the recovery of the tracers in other C and N pools during the post-drought period (Supplementary Fig. 9, Supplementary Table 4, $P > 0.05$).

## Discussion
Our findings on the reciprocal flow of C and N through plants and soil food webs shed new light on our understanding of how grassland management modifies the response of multitrophic networks to drought, with consequences for the key ecosystem processes they regulate. We found a stronger legacy effect of drought on fluxes of recent photosynthate below-ground in the intensively managed grassland, indicating impaired resistance of this process, likely through a decoupling of above- and below-ground interactions (Fig. 6). In contrast, few legacy effects of drought on below-ground fluxes of recent photosynthate were detected in extensively managed grasslands, indicating greater potential to buffer impacts of climate extremes on above- and below-ground interactions.

The legacy effect of drought on plant C assimilation and its allocation below-ground differed between the two grassland management types. Although drought tended to have a greater effect on soil moisture in extensively managed grassland (trend of higher soil moisture reduction and significant slower recovery to the soil moisture observed in the controls), the effect of drought on C transfer was stronger in intensively managed grassland, confirming our second hypothesis. This finding indicates that the direction of the change in C transfer was not biased by the magnitude of the effect of the drought on soil moisture. During the post-drought period in intensively managed grassland, and despite decreases in above-ground biomass, plant shoot C uptake increased and C transfer to roots, bacteria and Collembola decreased compared to controls. However, in extensively managed grassland, drought had no significant legacy effect on the plant biomass or C transfer below-ground (except to Collembola). The difference in the patterns of responses between grassland management types supports the idea that coupling between plant communities and the soil food web is key for the resistance and resilience of soil functioning to perturbations, such as drought, and confirms the importance of rhizodeposition as a key driver of the structure and function of the soil food web.

Our finding is consistent with recent reports of greater resistance and faster recovery of plants in abandoned grassland relative to

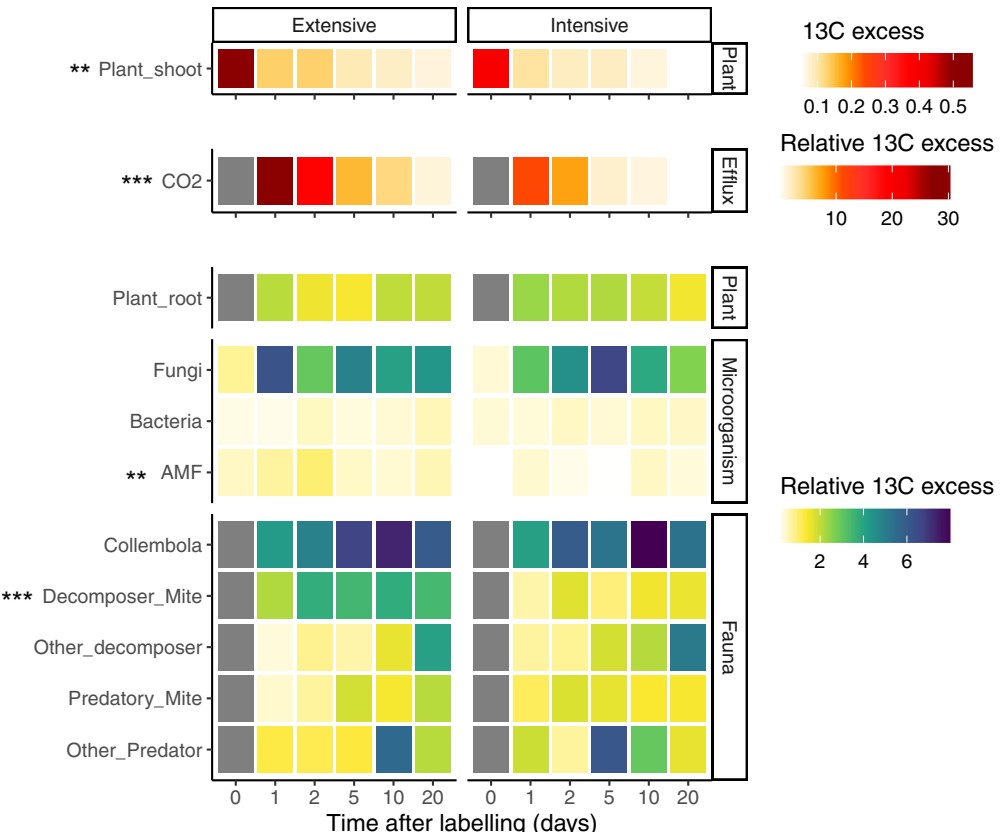

**Fig. 4 | Management impact on the ¹³C transfer in the plant-soil system.** Temporal dynamics of ¹³C in the plant shoot (upper panel), soil $CO_2$ efflux (middle panel) and below-ground pools (lower panel) following ¹³C pulse labelling (time 0) from the control plots ($n = 108$). The heat map shows the ¹³C-enrichment in different pools and fluxes in grassland under extensive (left) or intensive (right) management. The plant shoot enrichment is expressed in atom % excess; however, to be able to compare the flux of C from the plant to below-ground, all the other variables are expressed relative to the initial ¹³C fixed by the plants (relative ¹³C excess, %). Note difference in scales to visualise the relative enrichments. The difference between management and time was tested using two-sided mixed models. Results of the effect of the grassland management are reported with ***$P < 0.01$, **$P < 0.01$, *$P < 0.05$, see Supplementary Table 3 for detailed statistical results. Grey colour indicates no data available.

managed grasslands due to larger below-ground root and fungal networks in the former, which improves water access compared to intensively managed grasslands[28, 30, 32]. However, our previous work, which manipulated components of grassland food webs under controlled laboratory conditions, did not find any interactive effects of drought and food web composition on below-ground C flow and plant growth suggesting feedback effects may be of limited importance[23]. The modest duration, unique plant species and limited pool of soil faunal species used in that study may have precluded the development of tight interactions among plants and soil organisms that limited feedback effects. In the present study, grassland management influenced below-ground C flow and there were relatively small differences in food web composition, at least at the resolution measured (i.e., no differences in the biomass of microorganisms and only differences in biomass of some decomposers between the management regimes; Fig. 3 and Supplementary Fig. 8). We suggest that modification of the food web structure and composition at finer resolutions, or an enhanced efficiency of nutrient transfer within the soil food web, could drive feedbacks between plant and ecosystem functioning and their response to drought. Further work is needed to understand how shifts in the composition of complex food webs formed in the natural environment, such as those observed in the current study, feedback to plants over longer time periods and regulate ecosystem processes.

The broad classification of grassland in this study into extensive and intensive management regimes captures a suite of above- and below-ground properties. Although we measured differences in key properties between the grassland types (Fig. 3; Supplementary Figs. 4

and 6), we are unable to unequivocally separate proximate and ultimate effects of management intensification on below-ground C flow. There are many factors involved in grassland management, including fertiliser use, differences in grazing intensity and compaction associated with livestock and machinery use[33]. All these factors have potential to impact soil abiotic and biotic properties and vegetation (Fig. 3; Supplementary Figs. 4 and 6) and disentangling their impact on below-ground C fluxes in response to drought would require additional manipulation experiments in the field, which was beyond the scope of this study. Nevertheless, the greater uptake of C by plants in intensively managed relative to extensively managed grasslands in response to drought could be a compensatory effect following the release of the drought. Indeed, fast-growing plants, which dominate the plant community of intensively managed grasslands, typically have an ability to open their stomata more quickly when drought is released compared to slow-growing plant species[34], which dominate the plant community in extensively managed grasslands. Our results confirmed that intensive management promotes fast-growing plant species with lower root-to-shoot ratio. Typically, these species store resources in roots to facilitate regrowth after cutting, while extensive management promotes plants that invest in root growth, rather than storage, to access soil resources[30]. Despite large differences in plant biomass allocation, management intensity had no detectable effect on root ¹³C enrichment, indicating a similar rate of root C allocation of newly incorporated photosynthates in both management regimes. This contrasts with the general idea that slow-growing plants allocate more C to their roots[35], and potentially indicates a rapid transfer of root-¹³C

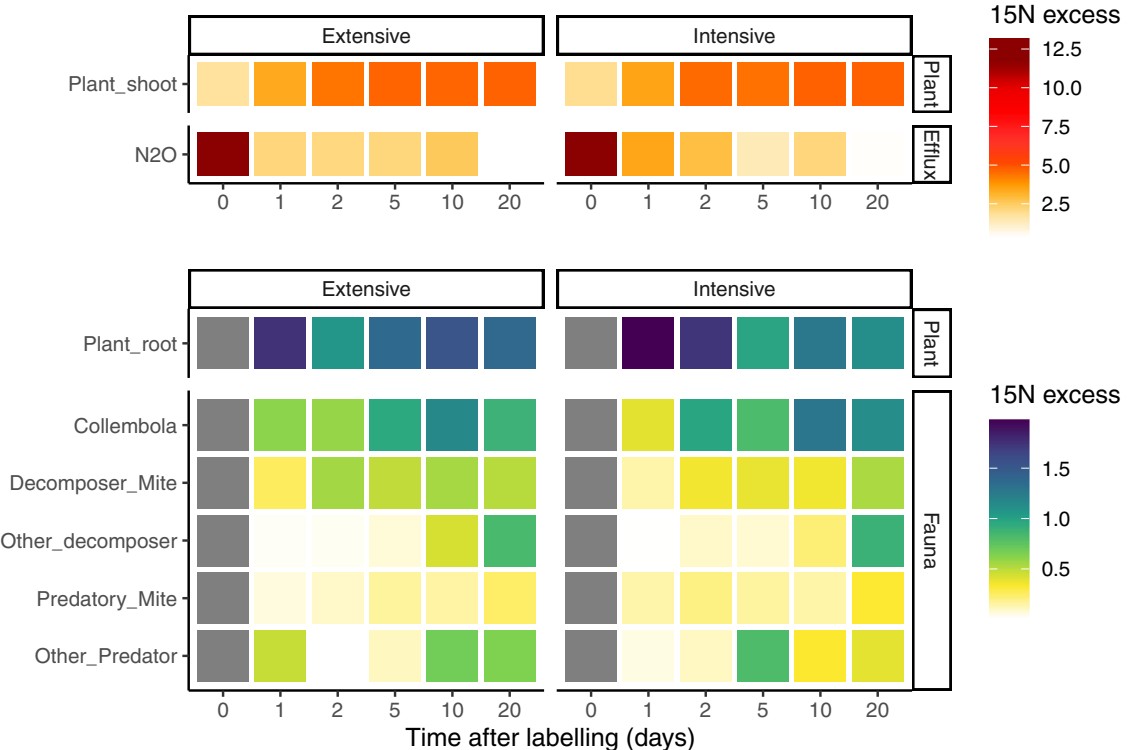

**Fig. 5 | Management impact on the [15]N transfer in the plant-soil system.** Temporal dynamics of [15]N atom % excess above-ground (upper panel) and below-ground (lower panel) in plant and soil fauna following [15]N pulse labelling from the control plots ($n = 108$). The heat map shows the [15]N atom % excess in the different groups in grassland under extensive (left) or intensive (right) management. Note difference in scales between the two panels as the enrichment of above-ground nitrogen pools and fluxes are substantially greater than any of the below-ground pools. The difference between management and time was tested using two-sided mixed models. There were no significant effects of grassland management: see Supplementary Table 3 for detailed statistical results. Grey colour indicates no data available.

to below-ground organisms, in line with plants with thick roots diverting C to collaboration with soil organisms[36, 37].

Fungi and Collembola were a major conduit of recent photosynthate-derived C; on average 5.8% of fungal C and 7.6% of Collembola C came from plant recent photosynthates at their enrichment peak. The greater [13]C-enrichment of non-mycorrhizal fungal PLFA compared to the AM fungal PLFA points to an important role of non-mycorrhizal fungi in channelling plant-derived C into the soil food web. This finding not only supports other recent findings that an important portion of plant derived C go to saprotophic fungi[38–41], but also may reflect the fact that saprotrophic fungi form a significant portion (20–66%) of microbial biomass in a grassland rhizosphere[42]. Moreover, it may reflect a rapid turnover of [13]C in AM fungi[43] and consequent loss of [13]C as respiration[44, 45].

Although the recovery of the [13]C tracer was greater in bacterial PLFA compared to AM fungal or general fungal PLFA (see Supplementary Fig. 9), the conversion of PLFA to biomass is higher for fungi, and this means the absolute amount of [13]C in bacterial biomass was less than for fungal biomass (11.8 nmol of the PLFA 18:2ω6,9 = 1 mg of fungal C while 363.6 nmol of total bacterial PLFA = 1 mg bacterial C). A greater proportion of plant-derived C was recovered in AM fungi in extensively managed compared to intensively managed grassland, despite similar biomass in the two systems. This pattern may be due to enhanced demand or efficiency of C uptake by AM fungi in extensively relative to intensively managed grasslands. Similar results have been found for non-mycorrhizal fungi during the course of nature restoration on abandoned arable land, where fungal biomass was not impacted, but enhanced efficiency of C uptake by the fungi was observed[40]. Our results confirm that fungi have a more important role in extensively managed than intensively managed grassland[14] and that

they promote the retention of recently assimilated C in soil[28, 38], at least during the time scale of this study.

Drought decreased the flow of [13]C in the intensively managed grassland through several components of the food web, in particular bacteria and Collembola. This could be explained by the fact that Collembola are more sensitive to drought than other soil fauna and often reduce their activity in response to drought[23] and bacteria grow quickly and are more sensitive to drought and other stresses than fungi[46–51]. Furthermore, some fungi, in particular mycorrhizal fungi, are able to create large hyphal networks that facilitate nutrient and water transfer over long distances, and indirectly benefit plants by exploring water-filled soil pores not accessible to plant roots[50, 52, 53]. Although unlikely, it is also possible that species producing extensive networks could have been influenced by conditions outside the experimental plots.

Our results show that soil $CO_2$ efflux is relatively less enriched in [13]C in the intensively managed compared to extensively managed grassland in control conditions. This finding indicates that in intensively managed grassland, soil activity (including roots and soil organisms) relies proportionally less on C derived from recent photosynthate compared to extensively managed grasslands. This finding can be explained partially by the fact that there is less root biomass (Supplementary Fig. 6), and so less C input below-ground, in intensive systems. Furthermore, our results show that there is less plant-derived C transfer to AM fungi and oribatid mites in intensively managed grassland, supporting the idea that intensive management modifies food web structure and decreases the flow of energy through AM fungi[24–27, 54]. Although our results showed only small differences in the food web structure, at least at the level of definition studied, we found a higher dispersion of the soil communities in the extensively

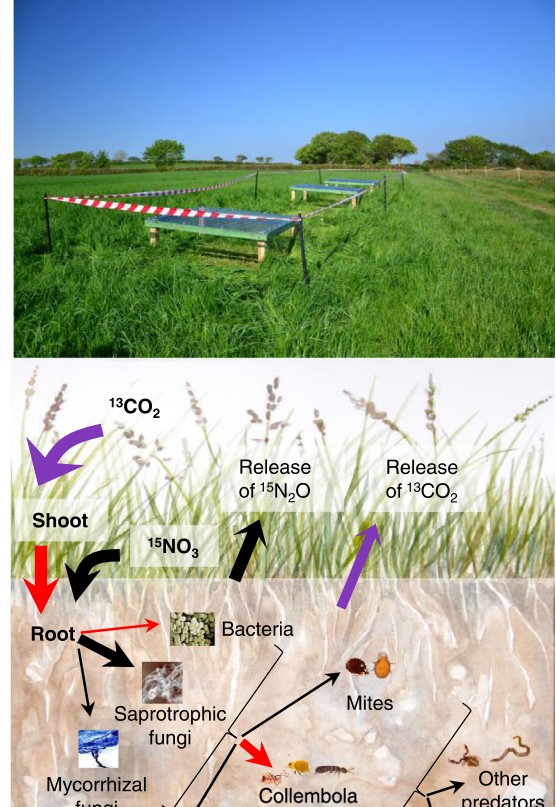

**Fig. 6 | Summary schematic on management and drought impacts on tracers transfer in the plant-soil system.** Schematic of the influence of grassland management (left panel extensive, right panel intensive) and drought (negative impact in red, positive impact in violet, and no significant impact in black) on [13]C uptake by plant shoot and transfer to roots, soil organisms and $CO_2$ efflux, and [15]N transfer to

$N_2O$ efflux. The size of the arrows corresponds to the relative proportion of [13]C or [15]N incorporated by the plant, soil organisms and released as soil efflux. Pictures by M. Chomel, grassland illustration by J. C. Bardgett. Note that we present only the main effects, so every arrow (except the $N_2O$ efflux) concerns [13]C transfer. The transfer of [15]N to soil organisms was not impacted by drought.

compared to intensively managed grassland. This result could indicate that intensive management promotes homogenisation of the relative abundance of the main soil fauna groups, potentially due to homogenisation of soil properties in space[55]. Rewetting at the end of the drought period led to an increase of respiratory losses of recent C from the ecosystem in intensively managed grassland. During drought, C pools may accumulate and become metabolically available for roots and microbes upon rewetting[32]. In addition, microbes can become potentially active after drought within hours upon rewetting[56]; they can resuscitate from dormancy[46,57], and depending on the duration of drought, they can start to regrow within one to several days[58-60].

Surprisingly, [15]N transfer to plants and soil organisms was mostly unaffected by grassland management and drought, suggesting that soil-driven processes of resource transfer are more buffered against perturbations compared to plant-driven processes. The production of $N_2O$ in soils is typically heterogeneous leading to hot-spots and hot-moments[61], and in extensive grassland there was marginally greater emission of $N_2O$ (which we attribute primarily to the nitrate reducing processes of denitrification and nitrate ammonification) immediately after the ammonium nitrate was injected into the soil and during the following day (Supplementary Fig. 7). This contradicts our first hypothesis and the general idea that intensive grassland management enhances the emission of $N_2O$ [62,63]. This is likely to reflect the higher C availability in extensively managed grassland providing the reductant for more sustained nitrate reduction[64]. Moreover, in extensive grassland, drought did not modify $N_2O$ efflux, but it decreased its

[15]N-enrichment compared to the control (in intensive grassland, there was a significant effect at day 1 only). Soil moisture has consistently been shown to be one of the most important parameters affecting soil oxygenation and thus determining $N_2O$ production rates[61,65], and may change the microbial source of measured $N_2O$[66]. Under drought, higher concentrations of $O_2$ in soil pores may favour ammonia oxidation of unlabelled $NH_4$ resulting in [14]N-dilution of the [15]N-$NO_3$ pool, and of the subsequent $N_2O$ emission.

Our findings show that intensive grassland management impaired the response of plant-derived C flow during a post-drought period, likely through a decoupling of above-below-ground interactions. The cause of this decoupling is uncertain, but likely involves a suite of interacting factors related to management regimes that impact biotic and abiotic properties. Moreover, our study demonstrates how the interplay between land management and climate extremes regulates bidirectional flows of nutrients through multitrophic networks. Surprisingly, grassland management and drought had little influence on the flow of added N to soil organisms. The transient nature of $N_2O$ fluxes and the variety of mechanisms contributing to its production prevent us from providing robust conclusions on its response to drought and management. In contrast to N cycle response, intensive management alters the ability of plants and soil organisms fuelled by recent photosynthate to buffer the response of C process to an extreme event. Our findings demonstrate that capture and rapid transfer of photosynthate through multitrophic networks is a key process for maintaining grassland resistance to drought: in the face of

increasing drought, intensive grassland management decreases C uptake and weakens the C sink of mesotrophic grasslands. Although the movement of C below-ground cannot be used as a specific indicator of soil health[67], it may be a good indicator of how systems perform in response to perturbations. This highlights the need for future studies on a range of soil and grassland types to examine trade-offs of grassland management intensities for various climate change scenarios (e.g. drought), which will identify resilient grassland management practices that can deliver sustainable food security in the face of climate change.

## Methods

### Field sites and experimental design

The field experiment was carried out in 2016 across a series of mesotrophic grasslands located in the Yorkshire Dales National Park, northern England (mean annual temperature 7.3 °C, mean annual precipitation 1382 mm). Three geographically distinct sites were selected, each with adjacent, paired grasslands (field) on the same soils and of similar topography but with different long term (>20 years) history of intensive or extensive grassland management (see Supplementary Fig. 1 and Supplementary Table 6 for more details). Extensively managed grasslands had not received any additions of inorganic fertiliser, were not cut for hay and had been grazed at low stocking densities by sheep or cattle (typically grazed at <1 livestock unit ha$^{-1}$ y$^{-1}$)[33]. Intensively managed grasslands received >100 kg N ha$^{-1}$ y$^1$ and were grazed at high stocking densities or for longer (typically grazed at 2-3.5 livestock unit ha$^{-1}$ y$^{-1}$), and were cut for hay once a year[33]. In general, plant communities of the extensively managed grasslands were species-rich *Anthoxanthum odoratum-Geranium sylvaticum* hay meadows (MG3 or subcategories), whereas plant communities of intensively managed grasslands were species-poor and classified as *Lolium perenne-Cynosurus cristatus* (MG6, MG7, and subcategories), according to the UK National Vegetation Classification[31]. The soils were all humose loamy brown earth (Cambisol, pH -5.5; 11.4% total C; 0.76% total N).

Most of the studies (72%) reviewed by Knapp et al. used a passive approach to reduce rainfall inputs based on a relative reduction in ambient precipitation amounts[68]. However, many of these studies were partial passive reduction all year round to simulate extreme precipitation years (reduction shelters), while our shelters reduced rainfall by 100% only in the summer to simulate extreme weather events (exclusion shelters), which are predicted to increase in frequency, intensity and duration by climatic models. In each field, an extreme drought event was simulated by placing passive rain exclusion shelters which reduced rainfall by 100%[68]. Three transparent roofs (1.5 m * 1.3 m) alongside delimited control plots were set-up for 60 days (17-19 of May – 17-19 of July 2016) designed to simulate a 100-year drought event[69] (See Supplementary Fig. 2 for set-up pictures). The sample design is then 3 sites * 2 management type (extensive/intensive) * 2 treatments (control/drought) * 3 pseudo-replicates for a total of $n$ = 36 plots. In each plot, a 40 cm metallic collar (hereafter "subplot") was inserted into the first 7 cm of soil before the treatment was initiated. At the centre of this collar a smaller plastic collar of 11 cm diameter was inserted and vegetation inside clipped to allow measurement of gas fluxes from the soil only. After the removal of the shelters, each plot (control and drought) received four litres of water to release the drought and encourage photosynthetic activity and the isotopic labelling started as detailed below.

### Measurement of soil properties

Continuous soil moisture measurements were made with SM300 moisture probes inserted in the soil up to 5 cm connected to GP1 data loggers (Delta-t, Cambridge, UK) in each plot at the site 5. Sequential soil moisture measurements were made with WET-2 sensor probes inserted in the soil up to 7 cm (Delta-t, Cambridge, UK). Soil pH was measured using standard methodology (Roberston et al., 1999) in a soil and water suspension with a pH metre (Seven2GO Mettler Toledo, Colombus, USA). For each plot a bulk sample was collected with a corer (4.2 cm diameter and 6.6 cm depth) for the determination of bulk density (Robertson et al., 1999).

### Isotopic labelling

We followed the movement of $^{13}C$ and $^{15}N$ into the plant shoots, through the soil community (microorganisms and mesofauna) and back to the atmosphere as $CO_2$ and $N_2O$ emissions across a period of 20-days (Fig. 1). Five hours after watering the plots, 130 ml of a solution of $NH_4-^{15}NO_3$ (Nitrate-$^{15}$N, 98 atom %; CK Isotopes Ltd, Leicestershire, UK) was applied directly as a single pulse into the first 7 cm of soil with a side-hole needle by injecting into 20 locations within the subplots ($n$ = 36 subplots). The quantity of N injected (equivalent to just 20 kg N ha$^{-1}$) was kept as small as possible to minimise potential artefacts associated with large nutrient additions, whilst still enabling quantification of the tracer in soil organisms. Three hours later, a composite soil sample was taken consisting of three cores (1 cm diameter * 7 cm depth) from each subplot. The following morning, vegetation in the same subplots were labelled with 99 atom % $^{13}C$-$CO_2$ (Sigma aldrich). We used an air-tight chamber constructed with a plastic bell cloche (approx. 20 L) equipped with 2 small fans to disperse the gas, and a rubber septum at the top to inject the gas (see Supplementary Fig. 2). Prior to the start of the $^{13}C$ labelling ca. 10:00 GMT, photosynthetic rates were measured using an infrared gas analyser (EGM-4, PP Systems, Hitchin, UK) to determine the timing of the $CO_2$ injections in order to limit excessive or deficient $CO_2$ concentrations in the chamber. During approximately 2-3 h, 25 mL of $^{13}C$-$CO_2$ were regularly injected through the septum for a total of 250 ml per subplot. The twelve subplots within a site (i.e. both extensive and intensive management regimes) were labelled at the same time. The three sites were labelled over 3 consecutive days for logistical reasons.

Immediately after the $^{13}C$-labelling, a small subsample of plant shoots (ca. 0.5 g), 3 small soil cores (1 cm diameter, 7 cm depth), and gas samples were taken from each plot. At 1, 2, 5, 10 and 20 days after the $^{13}C$-$CO_2$ pulse labelling (based on time course from previous studies[23, 39, 70]) gas samples were taken and a fifth of the soil from the subplots (metal ring) was harvested up to 7 cm depth (Supplementary Fig. 2). We refer to day 1 as being approximately 24 h after $^{13}C$ labelling and 36 h after $^{15}N$ addition. The holes created were filled back with sand to minimise gas exchange from the exposed surface and disturbance. Four supplementary cores outside the plots were taken per field to determine the $^{13}C$ and $^{15}N$ natural abundance signatures of each C and N pool.

A portion of the intact sample (corresponding to approximately 200 g of soil) was placed on Tullgren funnels over 7 days to extract mesofauna, collected in 70% ethanol and stored until further analysis. From the remaining sample, plant shoots were collected, and roots were separated from the soil by handpicking, and washed. Plants shoot and root samples were oven dried at 60 °C and the biomass was recorded. All samples were then homogenised and ground with a ball mill for analysis of total C and N content and $\delta^{13}C$ and $\delta^{15}N$ signatures using an elemental analyser (PDZ Europa ANCA-GSL, Sercon Ltd, Crewe, UK) coupled to a 20-20 isotope ratio mass spectrometer (Sercon Ltd, Crewe, UK). A subsample of the remaining soil was sieved, homogenised and freeze dried prior to the PLFA extractions.

### Soil microbial community analysis

Soil microbial communities were characterised by the extraction of the phospholipid fatty acids (PLFAs), according to Buyer and Sasser method[71]. Details of the method can be found in Chomel et al. (2019). The $\delta^{13}C$ signatures of individual PLFAs and their quantification were analysed by GC–C–IRMS using a GC Trace Ultra with combustion column attached via a GC Combustion III to a Delta V Advantage IRMS

(Thermo Finnigan, Bremen, Germany; Thornton et al. 2011). The internal standard 19:0 phosphati- dylcholine (Avanti Polar Lipids) added at the beginning of the extraction procedure was used for calculating concentrations. In summary, 36 PLFAs were identified in these samples, of which 20 microbial specific PLFAs comprising approximately 80% of the total concentration were used in subsequent data analysis. The fatty acids i15:0, a15:0, i16:0 and i17:0 were used as biomarkers for Gram positive bacteria; 16:1ω7, 18:1ω7, cy17:0 and cy19:0 were used as biomarkers of Gram-negative bacteria; and 15:0, 17:0 were used as general bacterial markers[72]. The fatty acids 10Me17:0 and 10Me18:0 were used as specific biomarkers of actinobacteria, and 17:1ω8c and 19:1ω8 were used as biomarkers of methane oxidising bacteria[34]. Gram-positive, Gram-negative, and general bacterial markers were summed to give total bacterial PLFA. 18:2ω6,9 was used as a marker of fungi (Bååth, 2003; Bååth and Anderson, 2003) and 16:1ω5 was used as a marker of AM fungi[73]. Although 16:1ω5 is used widely for estimating AM fungal biomass, its use can lead to uncertainties because bacteria can contribute to this pool[73]. However, in our case, the $^{13}$C-enrichment of this PLFA was very distinct from all bacterial PLFAs, which gave confidence in its use to estimate AM fungal biomass and its $^{13}$C uptake. The $\delta^{13}$C value of each PLFA molecule was corrected for the C added during derivatization using the Eq. (1).

$$\delta^{13}C_{PLFA} = \frac{C_{FAME} \times \delta^{13}C_{FAME} - C_{MeOH} \times \delta^{13}C_{MeOH}}{C_{PLFA}} \quad (1)$$

where $C_{FAME}$, $C_{MeOH}$, and $C_{PLFA}$ denote the number of carbon atoms in the FAME, methanol, and PLFA, respectively, and $\delta^{13}C_{FAME}$ and $\delta^{13}C_{MeOH}$ are the measured $^{12}$C/$^{13}$C isotope ratios of the FAME and methanol, respectively (methanol $\delta^{13}$C = −29.3‰). While the fungi are represented by only one PLFA, the bacterial community is represented by several PLFAs. To calculate an overall $^{13}$C-enrichment (atom% excess) of bacterial PLFA, the net $^{13}$C of all individual bacterial PLFA were summed and divided by the sum of the net C of all individual bacterial PLFA using the Eq. (2):

$$Atom\%^{13}C_{BacterialPLFA} = \frac{\sum(C_{PLFA_i} \times atom\%^{13}C_{PLFA_i})}{\sum C_{PLFAi}} \quad (2)$$

where $C_{PLFAi}$ is the amount of carbon from individual PLFA and atom% $^{13}C_{PLFAi}$ is the $^{13}$C-enrichment of the individual PLFA.

## Mesofauna

Individuals were counted and identified under a dissecting microscope to order and trophic group for Collembola[74] and Acari[75]. Other invertebrates were separated according to taxa and trophic group. The biomass of each order/taxa was estimated by measuring the average body length (mm) per field, using the formula by Caruso and Migliorini[76] for the mites and by Ganihar[77] for the other orders. The samples were further grouped into 7 main trophic groups in order to have sufficient material to analyse $^{13}$C and $^{15}$N: detritivorous Collembola, detritivorous mites (oribatid, astigmata and prostigmata mites), annelids, other detritivorous (detritivorous coleoptera, myriapoda and diptera larvae), herbivores (hemiptera and thysanoptera) predaceous mites (mesostigmata and predaceous prostigmata) and predaceous fauna (arachnida, chilopoda, predatory coleoptera and symphyla) (Fig. 1). Each of these groups was transferred into a tin capsule in 70% ethanol, oven-dried and weighed prior to analysis. All the samples were analysed for total C and N content and $\delta^{13}$C and $\delta^{15}$N using a Flash EA 1112 Series Elemental Analyser connected via a Conflo III to a DeltaPlus XP isotope ratio mass spectrometer (Thermo Finnigan, Bremen, Germany).

## Gas samples

Gas samples were taken by placing a 1.2 L dark chamber over the gas sampling core. Immediately after the closure of the chambers, and after 10, 20, and 30 min, 15 mL gas samples were taken from the headspace of the chamber using a gas-tight syringe fitted with an SGE syringe valve and transferred into pre-evacuated 12 mL gas-tight vial (Labco Ltd. UK). At the last time point, an additional 120 mL gas sample was taken and stored in an He-flushed, pre-evacuated 120 ml gas bottle fitted with Silicone/PTFE septa (Supelco) for $^{15}$N-N$_2$O analysis. CO$_2$ and N$_2$O concentrations were analysed on all samples in a gas chromatograph 7890 A GC (Agilent Technologies, USA) configured with a single channel using two detectors, an FID and a micro-ECD. The $\delta^{13}$C values of the samples from the last sampling point were analysed on a Picarro G2131-*i* Isotope and Gas Concentration Analyser (cavity ringdown spectrometer). The $\delta^{15}$N values of the samples from the last sampling point were analysed using a Sercon Ltd isotope ratio mass spectrometer following cryofocusing in an ANCA TGII gas preparation module (Sercon Ltd, Crewe, UK).

## $^{13}$C and $^{15}$N calculations

The isotopic concentration data was converted from $\delta^{13}$C and $\delta^{15}$N values (‰) to atom % excess $^{13}$C and $^{15}$N by subtracting the atom % $^{13}$C and atom % $^{15}$N of unlabelled controls from each enriched sample[78]. $^{13}$C- and $^{15}$N-enrichment is independent of the pool size and is indicative of the replacement of C or N from the pool by newly incorporated plant-derived $^{13}$C or fertiliser-derived $^{15}$N.

Although the plants were pulse-labelled with the same amount of $^{13}$C-CO$_2$, differences in photosynthetic and respiration rates caused the initial amount of $^{13}$C fixed to differ among subplots. To compare the relative transfer of $^{13}$C from the plant shoot into below-ground C pool (roots, microorganisms and soil fauna) in the different systems, the results were expressed as a percentage of the initial plant shoot $^{13}$C enrichment.

The net incorporation of the $^{13}$C or $^{15}$N tracer into the different carbon/nitrogen pools for each subplot was calculated using the Eq. (3)

$$^{13}C\ pool\ size = C\ pool \times atom\%^{13}C\ pool \quad (3)$$

where C pool is the amount of C in each pool (gC per subplot) and atom%$^{13}$C pool is the atom % excess of $^{13}$C of each pool. The same equation is used for the net incorporation of the $^{15}$N, replacing $^{13}$C and C by $^{15}$N and N. In order to evaluate the recovery of the stable isotopes in the system, we calculated the fraction of the isotope in the different compartments as the ratio of the pool size in a compartment relative to the total initial $^{13}$C pool size in the plant shoot or the total $^{15}$N pool size injected in the soil. These were calculated only for the relevant time point for which the enrichment was maximal: at day 1 for the plant and microorganisms and at day 5 for the soil fauna.

## Data analysis

All analyses were performed using R software (version 4.1.0. R core Team, 2017). To assess the role of land management in driving the soil properties and plant biomass, we performed a principal component analysis (PCA). Principal coordinates analysis (PCoA) was applied to estimate the soil community structure using Bray–Curtis dissimilarities based on Hellinger transformed community data[79]. A permutational multivariate analysis of variance (PERMANOVA)[80] was performed with 999 permutations to assess whether there were significant differences among grassland management. Distance comparisons in PERMANOVA can be sensitive to between-group differences in dispersions, so we used PERMDISP[81] to determine whether the dispersions of each group around their group centroid were significantly different from one another.

To analyse the effect of land management on plant and soil organisms biomasses we used a mixed model with management as

fixed effect and time and site as crossed random effects on data from control plots only[82] ($n = 90$).

To analyse the effect of land management on the $^{13}$C- and $^{15}$N-enrichment we used a mixed model with site as random effect and allowed different variance for each time point (with the function weights=varIdent(1|time)) from control plots only ($n = 90$).

To quantify drought impacts, and its interaction with grassland management, we used an effect size calculated as the log Response Ratio from each paired plots (logRR), which quantifies the proportional difference between mean $^{13}$C-, $^{15}$N-enrichment or biomass in control and drought conditions[45]. On a log scale, an effect size of 0 means there is no difference, a positive value means a positive effect of drought, while a negative logRR means a negative effect of the drought. To increase robustness, and because the trend of the response variable over time was relatively similar, we used the average value of each replicate across all sampling dates in our calculation of logRR. We then performed a mixed model with management as fixed effect and plot within site as random effect and used the 95% confidence interval of predicted means to specify if the logRR was different from 0 ($n = 18$).

### Reporting summary
Further information on research design is available in the Nature Portfolio Reporting Summary linked to this article.

### Data availability
The data generated in this study, including biomass, stable isotope enrichment and fauna abundance data, have been deposited in Figshare and are available at https://doi.org/10.6084/m9.figshare.21395013.v1.

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

## Acknowledgements

This study was supported by a consortium grant from the Natural Environment Research Council (NERC) Soil Security Programme led by R.D.B., with component grants NE/M017028/1 to R.D.B. and F.d.V., NE/M01701X/2 to D.J. and E.M.B. and NE/M017036/1 to T.C. and M.E. We are very grateful to the landowners and farmers for allowing us to perform our experiment and sample their fields. We also thank Lucy Frotin and Juliette Papelard for help in the laboratory and field, H. Stott for help with isotope analyses, L. Harrold for plant and $N_2O$ isotope analysis, B. Thornton and G. Martin for the PLFA and soil fauna isotope analysis, T. Balanant for the map figure, and C. Collins for his support as coordinator of the NERC Soil Security Programme.

## Author contributions

R.D.B. initiated and gained funding for the project with D.J., T.C., F.d.V., E.B. and M.E. M.C., J.L., F.d.C., R.D.B., D.J., M.E., T.C., F.d.V. and E.B. conceived and designed the experiment. M.C., J.L., F.d.C., N.A., J.R. and M.M. set-up the main experiment, M.C. and N.A. performed the pulse labelling and field sampling. M.M. identified soil fauna, M.C., N.A. performed laboratory analyses. M.C. statistically analysed the data and led writing the manuscript in close consultation with D.J., T.C. and R.D.B. and with discussions and contributions from all authors.

## Competing interests

The authors declare no competing interests.
