## [Peer Review File · Nature Communications]

Reviewer comments, first round review -

Reviewer #1 (Remarks to the Author):

Chomel et al. performed field experiments to investigate the effects of drought on extensively and intensively managed grasslands, respectively. This is a highly topical question with climate change leading to more frequent drought events and the intensification of agriculture due to a growing world population. The bulk part of the data is presented in the form of biomass estimation (PLFA, mesofauna approach) and gas fluxes in conjunction stable isotope experimentation. I particularly enjoyed the visualization of temporal dynamics of stable isotope incorporation into plants and soil organisms. While the authors find some differences in the response of stable isotope fluxes in response to drought, the presented data does not give enough evidence to justify a generalized conclusion on which management type has higher resistance and resilience to drought. The variability among the response ratios is very high and only few parameters showed a significant effect. At times the interpretation of the data seems to be a bit overreaching.

My main concern revolves around the comparison of drought treatments for both management types. The extensively treated plots had approx. 10% lower soil moisture for most of the drought period (see Fig S1) which at such low levels of total soil moisture will have a huge impact on the activity of soil food web. Think about connectivity of biological interaction, a soil with 20% soil moisture (intensively managed) will provide proportionally more water films, connected pores, allow for motility and interaction within the soil architecture, than a soil with only 10% soil moisture. The differences in soil moisture (and soil compaction!) will certainly play an important role in the response to the applied drought treatment. Of course such treatments like drought are very difficult to control in field experiments and while I am not saying the experimental design was flawed, these differences render the comparison of pairs (extensive vs intensive) under drought very difficult. In addition, isn't it plausible to argue that the higher grazing by cattle in intensively managed plots led to soil compaction and that this would be responsible for the apparent "decoupling of above- and belowground interactions"? Thus, I strongly advise the authors to critically discuss these background parameters in a revised version and what their effect could have been. On another note, comparing the response to drought to "historic" influence would certainly benefit from more measured variables that describe the soil environment. I assume that the soil nutrient status and availability (DOC, TON) would be largely different between these plots. The detail provided in the methods does not allow for the work to be reproduced. The presentation of the methods is incomplete as some of the measurements are not even described in the text (e.g. pH, bulk density). To increase the understanding of the experimental setup I suggest to add a photograph or schematic drawing that also shows the location and number of samples taken. In the current version, I am having a hard time understanding where samples were taken per plot, how many samples in total per treatment, how many replicates, what depth the soil samples were taken from (see compaction comment above), etc. Also it does not become clear how much soil was taken overall per composite sample, how much soil was used for the individual measurements, and so on. The study is not reproducible at all the way it is presented now. In general, the presentation seems to be a bit sloppy. For example, Figure 6 is mentioned in the text but cannot be found in the document.

Specific comments:

L85-90: The hypotheses seem to be a bit random, it does not become clear why C losses would decrease if more C was transferred from plant to AMF. Also I believe that the hypotheses need to be discussed more prominently in the discussion section.

L43. fluxes that occur belowground or fluxes into the belowground parts? please clarify

L42-44: If findings "indicate" decreased fluxes it is a bit far-fetched to state conclusions on the resistance of the given process?

L46: Overreaching: yes, belowground carbon allocation should represent a key process for resilience was not tested in this study. So how can a conclusion on resilience been drawn here?

L69: I am not sure we are too curious as to how the relative abundance of fungi and bacteria modulate transfers of C from plants to belowground pools. Why the relative abundance?

L70: belowground or below ground

L108: intensification also meant more cattle per ha⁻¹. So what about soil compaction?

L116: Please provide an international classification e.g. based on the World Reference Base for Soil Resources

Fig1: the figure suggests there are no trophic interactions between Microorganisms. What about bacterivore bacteria? What about nematodes and acanthamoeba feeding on bacteria and fungi? Thus I believe that the depiction of the expected flow of stable isotopes through the food web is a bit simple.

L141: It does not become clear how many samples and replicates it was. Also where in the plots were these taken?

L152: At what depth were these samples taken? How did the authors make sure that the sample received label in the end? Were those rhizosphere samples?

FigS1: why did the soil moisture decrease more strongly in the extensively managed plots as compared to the intensively managed? Also it seems that the resistance of soil moisture to drought is higher in intensively treated plots.

L161: How were plant samples prepared for measurement of isotopic enrichment? Were samples homogenized? Same applies for soil samples

How much soil was taken overall (in grams) and how was it divided for the methods? It seems that 3 x 1cm cores do not yield a lot of soil material to begin with, how sure are the authors that the stable isotopes retrieved in those samples are representative of the soil food web? Also since the biomass of mesofauna was calculated

L284f: Please provide the values for the different parameters in a supplementary table. L116 says that soil had a similar pH of 5.5 but here you say that the pH was higher in intensively managed soils.

Fig2: Please provide a more extensive caption of what it shown. It does not become clear what "ento_bg" and other labels could stand for. Also, what are the PLFA data in there? Are they aggregated into groups? Also are the treatments drought and non-drought included as well as the sampling days?

L.301: I guess that ind.m⁻² should stand for individuals per square meter but please introduce the abbreviation beforehand once. Also it does not make sense to have 3 digits for individuals? Also it does not become clear if these data come from the control plots and are averaged over replicates as shown in FigS4.

L301-303: "Collembola was the most abundant group..." But of what?

L307: PERMDISP

L317: "13C pulse labelling of plant shoots". It occurs to me that the atmosphere was labeled where the plant would fix the labeled CO₂. Thus, no labeling of the plant shoots was performed. Is this correct?

L334: "Plant leaves in..." There is no mention of plant leaves in the figures and the methods so far. Were leaves measured instead of shoot material? Again describe in the methods

L344: True but the label found in these groups were also much lower than in other groups. Especially for AMF and bacteria, the incorporation of label is very low. Why could this have been the case?

Fig. 6 seems to be missing

L. 410f: The authors found an effect of drought on grassland management types, and this effect differed with regard to plant C assimilation and allocation belowground. However, how can the authors be sure that these effects are not simply due to the plant biodiversity and/or the soil moisture?

L456: I do not see how this statement is justified by the data presented. There is not causal relationship between soil activity and photosynthetically derived C studied here. The data does not allow the authors to judge on the total activity of participants within the system from stable isotope tracing as performed here.

Reviewer #2 (Remarks to the Author):

This study addresses the interaction between drought and management intensity in grasslands, monitoring ^{15}N and ^{13}C stable isotope signature through several levels of the soil food web over time. It is a very ambitious experiment that involves measurements in three paired sites over 20 days and a large array of scientific expertise. It is a very solid piece of work.

I suggest clarifying the way knowledge gaps are presented, since for the moment they are described in sequence with little integration: 1) how shifts in food web structure and f:b ratio modulate C transfer to the soil and plant nutrient capture(L69); 2) need for a multitrophic perspective to address the response of C and N losses to perturbation (L76); 3) response of rhizodeposition to changes in land use (L82). They also suffer from syntax issues, especially when listing items (L83 soil C and N cycling and sequestration; L78 the responses of processes of C and N loss to perturbations, such as drought; L79 awareness of the importance of rhizodeposition as a driver of... and the structure and functions of...; L89 incorporation of ... into... and CO_2 and N_2O fluxes...).

Improving the structure is not only important to clarify the gaps, but also to specify where the study fills them, and get back to that in the discussion.

The f:b ratio deserves some sort of introduction, as it appears out of nowhere. Further, I am not sure that "shifts in bacterial and fungal energy channels" (L66) is clear for all readers of this journal, please explicit or rephrase.

I assume that the authors chose to leave it out of the scope of the study, but from a topological point of view, I am left wondering how soil food web structure and complexity is affected. Are the nodes and connectance modified, is there any rewiring? I think it is relevant to address this structural point, because it should be related to the fluxes of C and N that are the focus of the study.

Interaction between management intensity and drought is the heart of this study in my view. The results start with a focus on management and drought, the interaction with drought appearing in part 4. I think it would be nicer to explicitly state in the results that the interaction was significant before diving into the different responses under different management regimes, to keep a focus on the interaction rather than on the management treatment. This suggested structure is actually followed in the discussion, so restructuring the results would fit better with it.

I enjoyed the discussion, it is concise and to the point. Several aspects could still be improved. 1) It tries to explain everything at times, which is expected given the impressive dataset, but could be reduced. For example, L459 less C input belowground was not measured yet presented as such. 2) It sometimes opens arguments that are not resolved: L448 what timeline are we considering for the C retention? L427-4299 what is the authors' position considering the apparent contradiction?

The conclusion goes for generalisation, yet it is a difficult position to hold regarding CO_2 and N_2O emissions as they were measured in this study. N_2O emissions are typically very transient, and would have to be measured right after the rain events that occurred at the end of the experiment to catch a bump. It looks like it was the case on day 10, was it significant? More generally, how robust is the generalization of N_2O and CO_2 emissions response based on half a dozen point-in-time measurements over 20 days, especially when relating to generalisation of N_2O production L472?

Given the large dataset, could the authors identify in their study, and propose, a marker variable for specific ecosystem services? This could simplify the endeavour they propose in the last sentence of the conclusion.

Other comments

L35 photosynthates

L41 "However" contrasts N transformations against C transfer, which is a bit awkward in itself, and this sentence about N is also lost in the middle of a section in C transfer.

L43 disrupted, L45 provided
L35, 46, 47 photosynthates
L45 "impacts to drought" please rephrase
L53 Shifts in soil food web structure both regulate the stability of soil functions and impair their resistance. This is awkward wording.
L55, 45 I don't understand why "ability to buffer" is used instead of "resist to" as in stability theory.
L59 "These drivers" please explicit
L66 "including..." does not fit in the sentence
L150 change to consecutive
L363 change in function to as a function
L373 change its to ¹³C
L462 change to through
L468 change to marginally
L452-454 this information is general and its location does not participate in driving the discussion. Please restructure the entire paragraph L449-454.

Figures: please explicit variable names throughout the manuscript and supplemental, correct superscript/subscript for isotopes/gases and scientific notation of units. Please include units in all figures, even supplemental (S1, S3).

Fig. 7 is actually Fig. 6

Fig. S3 requires units and change efflux to efflux rate

Fig. S5 explicit legend

Fig. S7 change stock to pool size, along time to over time. Explicit that the numbers at the top are days after labelling.

Reviewer #3 (Remarks to the Author):

The authors describe an impressive experiment in which they manipulate drought in two types of grassland communities – extensively and intensively managed – and quantify carbon and nutrient transfers throughout the entire food web. Their results suggest that intensive grassland management has a negative influence on the diversity of the belowground food web (lower dispersion) and the resistance of carbon transfers belowground following drought, which they link to this disrupted food web. Drought led to reduced aboveground biomass in intensively managed grasslands but not extensive grasslands and the isotopically labeled carbon was concentrated aboveground with reduced transfers belowground in the intensive grassland. The paper is the first of its kind that I have read and was well written. However, there are some issues that I have with the methods and interpretation of results that need to be resolved prior to publication.

First, the authors should emphasize that they are measuring grassland community resilience or recovery from drought. No measurements were made during the drought treatments but rather in the days following re-watering.

Second, the 20-year management practices have led to significant changes to above and belowground community composition. Obviously management has caused these differences, but is there a way to disentangle the specific effects of community change on the C and N flows you describe?

Third, the introduction could use a bit more detail on how isotopically labeled C and N have previously been used to answer similar questions.

Finally, the interpretation of C¹³ results is puzzling to me as plant C¹³ is often a metric of water use efficiency. C¹³ naturally occurs in the air and leaves/shoots with a high concentration of C¹³ generally reflect high water use efficiency as stomata close and discrimination against the heavier isotope decreases. Do the differences you observe between these two grassland types in terms of C¹³ accumulation simply reflect differences in water stress? For example, greater C¹³

accumulation was observed in the intensive grassland during drought which experienced greater biomass reductions during drought. I'm unfamiliar with the methods used for concentrating gas around the plant tissue, but my assumption is that this would also influence leaf physiology simply by increasing ppm of CO₂ around leaves which leads to greater photosynthesis and eventual stomatal closure (leading to increased C₁₃ accumulation). Is the concentration of C₁₃ in the gas samples high enough that discrimination can be ignored and all carbon taken up is C₁₃? Are some of these issues resolved by comparing the C₁₃ in labeled to unlabeled controls? Either way, some discussion of how stomatal functioning influences vegetation C₁₃ accumulation is warranted.

Specific comments:

Line 41: change to lowered the rate of reduction

Line 111: Extensive fields were not cut for hay?

Line 117: Here would be a good place to mention sample design. From my reading, you have n=6 plots (3 control, 3 drought) in each field per site, correct? So, a total n=9 for the following treatments: "drought intensive", "drought extensive", "control intensive", "control extensive". But really, your level of replication is site, so n=3 for each?

Line 117: Also, note that these roofs were 100% exclusion. Most drought shelters are passive reductions, making this unique in that respect. See Knapp et al. (2017). Pushing precipitation to the extremes in distributed experiments: recommendations for simulating wet and dry years. *Global Change Biology*, 23(5), 1774-1782.

Line 137: Change to "5 hours after watering the plots"

Line 140: You injected the annual total N in one go? Is that appropriate? Also, this is significantly lower than the amount of N that the intensive grassland usually received correct?

Line 144-145: Could you include an image of this set up?

Line 148: Does this amount of labeled carbon increase ppm of CO₂ surrounding the plant? This can lead to increased photosynthesis followed by stomatal closure in some plants if so. Additionally, does this treatment completely replace natural C₁₂-CO₂ in the chamber with labeled C₁₃-CO₂? Or do the plants still discriminate against the heavier isotope.

Figure 2: Are only the significant variables shown in the PCoA? Also, I recommend making the labels clearer or including the meaning of the abbreviations in the figure legend rather than the supplementary material.

Line 301: Consider re-wording the start of this paragraph. Needs a topic sentence or lead-in to what "group" you're talking about. "In terms of soil community composition,..."

Line 283: Yes, at least for control plots.

Line 284-286: This is not clear to me from Figure S7. Can you show a summary stat across treatments? A more detailed description of what high C₁₃ or N₁₅ implies would also be nice.

Line 306: This dispersion result is particularly interesting as a metric of diversity and should be discussed more in the discussion.

Line 369-370: Does the lack of C₁₃ enrichment reflect continued drought stress and the plants not opening stomata to take in C₁₃? Stomatal closure varies dramatically between species

Line 411: No drought impact on aboveground biomass in extensive grassland is notable and should be mentioned here. Perhaps the drought was not severe enough

Line 453: This is a good point – should be specific here about how these biota could access areas

outside of the drought shelters as the soil was not hydrologically isolated from the surround soil matrix.

Line 487: Doesn't this go against your recent findings (Chomel et al. 2019) that drought acts independently of trophic interactions to influence carbon storage aboveground and flows belowground? Perhaps discuss the differences between these two papers in greater detail.

Figure 7. This is a very helpful figure but is never cited within the text. Notably, drought had a positive influence on C13 storage in tissues. How do you tease apart radiolabeled carbon 13 that you added from natural carbon 13 that is known to accumulate in drought stressed plants (particularly C3 plants) as stomata close and discrimination of heavier isotopes is reduced? Is this determined from the difference between non-labeled control tissue? Also, in figure 7 it is difficult to determine whether an arrow in the food web is referring to C or N transfers, yet these were different in their responses to drought.

Figure S1: There seem to be clear differences in the effectiveness of the shelters between extensive and intensive. Wasn't the point of watering to remove the effect of drought? In which case, the plots should have similar soil moisture after shelters are removed. At what depth was soil moisture measured?

Reviewer #4 (Remarks to the Author):

The authors present an interesting field study on the impacts of land management on soil food web responses to drought, with a focus on fluxes of C and N among plants, microbes and microfauna. In this study, extensively and intensively managed grassland sites were first exposed to a long period of summer drought, after which C and N fluxes were studied using isotope tracing. Most importantly, the authors showed that after drought, carbon transfer to the belowground system was limited in intensively managed systems but not in extensively managed systems. The study thereby shows another clearly negative implication of agricultural intensification on ecosystem functioning. I agree with the authors that this is an important finding. However, I do have a number of comments regarding the data analysis and presentation of the results, as well as some other suggestions to further improve the manuscript.

General comments

1) My first comment relates to Figure 2 and the corresponding PERMANOVA analysis. It is unclear to me which samples exactly are represented by the individual dots in the figure, as based on the experimental design, I would expect single soil community composition measurements per (sub)plot (18 per management treatment/36 in total). However, there are more samples shown in the PCA plot. To avoid pseudoreplication, I would suggest to only incorporate a single sample per plot (as done in Fig. S2), at least when the goal is to just examine the effect of management on community composition. However, I think it would also be insightful to distinguish between drought and non-drought plots in these analyses, as drought will likely have caused a shift in community composition, the strength of which may have depended on management.

2) The authors used t-tests to analyse whether drought positively or negatively affected nutrient transfers to different response groups. However, with these analyses differences in drought effects between intensively and extensively managed fields are not explicitly tested, while in my opinion these comparisons are also important. Moreover, by using t-tests it's not possible to incorporate among-site variation, which may have obscured treatment effects. Therefore, I would suggest to analyse the log response ratios using mixed models as well. To understand whether the response ratios are significantly different from zero, confidence intervals of predicted means can be examined.

3) While I understand that the used methodology limits the number of sites that can be examined, I think it would be good to acknowledge that the study only included a single soil type, and that studies in different soil systems are needed to test the generality of these results.

4) To improve accessibility to a wider audience, I would recommend the authors to expand the description of the examined food web. I think Figure 1 shows this food web well, but in the text the different food web components are not explicitly mentioned. Furthermore, key components of soil food webs, most notably protists and nematodes, have not been analysed, I assume due to methodological limitations. I think this is worth some lines of discussion.

Specific comments

L59: Please specify 'these drivers' in order to further clarify the goal of the study in this sentence.

L63-64: AMF also are soil biota, so perhaps write: '...decrease the abundance and diversity of soil biota, including arbuscular mycorrhizal fungi, and induce shifts in soil microbial community composition.'

Given that your study has a multi-trophic perspective and includes more than just microbes, it would be good to expand the second part of the sentence by highlighting that also non-microbial groups in the soil community are affected (e.g. micro-arthropods).

L64-68: I suppose 'including shifts in the relative abundance of bacterial and fungal energy channels' refers to 'soil food webs associated with agricultural intensification', but this is not completely clear. Do you mean 'changes in soil food webs associated with agricultural intensification'?

L106: 'and of similar topography' refers to the 'three geographically distinct sites', right? In that case I would suggest to write 'three geographically distinct sites of similar topography were selected'.

L270-272: If I understand it correctly, the drought treatment was not included as a fixed effect in these models? I think it would be insightful to do this, even when drought effects are separately presented in the analyses of response ratios.

L273-280: It would be good if the calculations done to produce Fig.S5 are also explained here. Furthermore, I assume that adjacent control and drought plots were used to calculate independent replicate logRRs, but it would be good to specify this.

L301: 'most abundant group of mesofauna'

Caption Fig. S5. It's unclear on which calculation the presented metrics are based. I thought they would be purely based on measured (PLFA-based) biomass, but I got confused by L70 ('on the ^{13}C transfer'). Could you please clarify this here and in the methods?

REVIEWER COMMENTS

Reviewer #1 (Remarks to the Author):

Chomel et al. performed field experiments to investigate the effects of drought on extensively and intensively managed grasslands, respectively. This is a highly topical question with climate change leading to more frequent drought events and the intensification of agriculture due to a growing world population. The bulk part of the data is presented in the form of biomass estimation (PLFA, mesofauna approach) and gas fluxes in conjunction stable isotope experimentation. I particularly enjoyed the visualization of temporal dynamics of stable isotope incorporation into plants and soil organisms. While the authors find some differences in the response of stable isotope fluxes in response to drought, the presented data does not give enough evidence to justify a generalized conclusion on which management type has higher resistance and resilience to drought. The variability among the response ratios is very high and only few parameters showed a significant effect. At times the interpretation of the data seems to be a bit overreaching.

We thank the reviewer for their comments. We have now clarified throughout the manuscript precisely what was measured, i.e., the response of C and N fluxes during a post-drought period, which captures aspects of both resistance and resilience following their strict definition (see response to comment L46 below).

To improve the statistical analysis of the response ratio and explicitly test the differences in drought effect between management regimes, we have performed a mixed model instead of t-tests. In this way, we can incorporate among-site variation and from the confidence intervals of predicted means from the model we can understand whether the response ratios are significantly different from zero. We are hoping that these changes, which improve the robustness of the analyses, better justify our conclusion about differences between grassland management. Further, we have toned down the interpretation of the data in the discussion and conclusion.

My main concern revolves around the comparison of drought treatments for both management types. The extensively treated plots had approx. 10% lower soil moisture for most of the drought period (see Fig S1) which at such low levels of total soil moisture will have a huge impact on the activity of soil food web. Think about connectivity of biological interaction, a soil with 20% soil moisture (intensively managed) will provide proportionally more water films, connected pores, allow for motility and interaction within the soil architecture, than a soil with only 10% soil moisture. The differences in soil moisture (and soil compaction!) will certainly play an important role in the response to the applied drought treatment. Of course such treatments like drought are very difficult to control in field experiments and while I am not saying the experimental design was flawed, these differences render the comparison of pairs (extensive vs intensive) under drought very difficult.

The reviewer is correct that the intensity of the drought cannot be controlled in field experiment and could differ between pairs of fields under intensive or extensive management and between the sites. The supplementary Figure 1 (now supp. Fig 3) showed continuous measurement of the soil moisture in two paired plots (Control-Drought in Intensive or Extensive) in one site only. To complement this figure, we have added a new supplementary figure which represents the averaged soil moisture of all plots in the 3 sites at sequential time points (supplementary Fig.4). Although there is the same trend (i.e. a stronger effect of the drought shelter on soil moisture in the extensively managed grassland compared to

intensively managed grassland), this is not significantly different. However, we can confirm that there is a quicker recovery of the soil moisture from droughted plot to control levels in intensively managed compared to extensively managed grassland. Furthermore, although the drought tended to have a greater effect on soil moisture in the extensively managed grassland, the effect of the drought on C transfer was stronger in the intensively managed grassland. This finding indicates that the direction of the change in C transfer was not biased by the magnitude of the effect of the drought on soil moisture.

We added this statement in the discussion section, see lines 288-293:

“Although drought tended to have a greater effect on soil moisture in the extensively managed grassland (trend of higher soil moisture reduction and significant slower recovery to the soil moisture observed in the controls), the effect of drought on C transfer was stronger in the intensively managed grassland. This finding indicates that the direction of the change in C transfer was not biased by the magnitude of the effect of the drought on soil moisture..”

In addition, isn't it plausible to argue that the higher grazing by cattle in intensively managed plots led to soil compaction and that this would be responsible for the apparent “decoupling of above- and belowground interactions”? Thus, I strongly advise the authors to critically discuss these background parameters in a revised version and what their effect could have been. On another note, comparing the response to drought to “historic” influence would certainly benefit from more measured variables that describe the soil environment. I assume that the soil nutrient status and availability (DOC, TON) would be largely different between these plots.

Our experiment was designed to test how typical extensive and intensive grassland management regimes, encompassing all the indirect changes associated with these land uses, modify the resistance of soil fauna and soil function to a drought perturbation. We did not set out to identify which specific management factors contribute to the responses detected, which would require additional manipulation experiments to disentangle all the proximate and ultimate effects of management on soil properties; this is beyond the scope of this study, which was based on comparing paired grassland sites under real world management regimes. However, we certainly agree with the reviewer that there are multiple factors that likely varied or co-varied with grassland management, which we now consider in more detail. Notably, we summarise in the PCA (supplementary Fig 6) data on pH, water holding capacity, plant community composition, bulk density (used as a proxy of soil compaction), soil mineral N concentrations, and total C and N to demonstrate how plant and soil properties vary with grassland management. Bulk density data is now added in the supplementary Table 6 as well. Moreover, we now discuss the potential role of these factors in our discussion, while acknowledging that we are unable to explicitly disentangle the role of individual components of management and their impacts on plant and soil properties. We have added the following text on lines 323-332: *“The classification of grassland in this study into extensive and intensive management regimes captures a suite of above- and below-ground properties. Although we measured differences in key properties between the grassland types (Fig 3; supplementary Figs. 4 and 6), we are unable to unequivocally separate proximate and ultimate effects of management intensification on below-ground C flow. There are many factors involved in grassland management, including fertiliser use and differences in grazing intensity and compaction associated with livestock and machinery use³³. All these factors have potential to impact soil abiotic and biotic properties and vegetation (Fig 3; supplementary Figs. 4 and 6) and disentangling their impact on below-ground C fluxes in*

response to drought would require additional manipulation experiments in the field, which was beyond the scope of this study.”

We agree that nutrient availability data would have been useful, however, due to technical issues we do not have data on DON and TOC, but only on total C and N, along with data on a comprehensive range of other key soil properties.

The detail provided in the methods does not allow for the work to be reproduced. The presentation of the methods is incomplete as some of the measurements are not even described in the text (e.g. pH, bulk density). To increase the understanding of the experimental setup I suggest to add a photograph or schematic drawing that also shows the location and number of samples taken. In the current version, I am having a hard time understanding where samples were taken per plot, how many samples in total per treatment, how many replicates, what depth the soil samples were taken from (see compaction comment above), etc. Also it does not become clear how much soil was taken overall per composite sample, how much soil was used for the individual measurements, and so on. The study is not reproducible at all the way it is presented now. In general, the presentation seems to be a bit sloppy. For example, Figure 6 is mentioned in the text but cannot be found in the document.

We have added more details in the methods section about sample numbers, and added a supplementary method section for the pH, soil moisture and bulk density measurements. We added two supplementary figures as suggested: supplementary Fig.1 with a map of the experimental sites and aerial view of the paired sites and supplementary Fig.2 with some pictures of the experimental set-up and soil sampling area. We further clarified the information about plot numbers in the text (line 476): *“The sample design is then 3 sites * 2 management type (extensive/intensive) * 2 treatments (control/drought) * 3 pseudo-replicates for a total of n = 36 plots”*.

Specific comments:

L85-90: The hypotheses seem to be a bit random, it does not become clear why C losses would decrease if more C was transferred from plant to AMF. Also I believe that the hypotheses need to be discussed more prominently in the discussion section.

The background of the hypotheses is set in the introduction with the current state of the art and published literature, but we agree that this was not clear. To address this concern of the reviewer, we have re-written the introduction to provide a strong case for the hypotheses and revised our hypotheses as follows (lines 128): *“Here, we experimentally investigated how grassland management modifies the transfer of recent photosynthates and soil N through plants and the soil food web during a post-drought period. Because intensive management modifies plant and soil communities, we expected intensive management to disrupt the coupling of C flow from plants to mycorrhizal fungi and the soil food web leading to: a) greater soil C and N losses as CO₂ and N₂O, because we expected shifts to bacterial-dominated food webs that are linked to faster rates of nutrient mineralisation; and b) a greater legacy effect of drought on C and N fluxes because we expected that intensive management decreases the stability of the soil food web and its ability to resist drought.”*

L43. fluxes that occur belowground or fluxes into the belowground parts? please clarify

L42-44: If findings “indicate” decreased fluxes it is a bit far-fetched to state conclusions on the resistance of the given process?

This sentence has been removed

L46: Overreaching: yes, belowground carbon allocation should represent a key process for resilience but resilience was not tested in this study. So how can a conclusion on resilience been drawn here?

Resistance to disturbance and the speed of return (resilience) are the two components of ecosystem stability as described by Pimm (1984), McNaughton (1994) and Loreau et al. (2002). Resistance is commonly defined as the ability of a system to withstand a disturbance, while resilience describes a trend towards pre-drought conditions (the directionality and speed of return) (Griffiths & Philippot 2012, Vilonen et al 2022). In this study we are describing ecosystem response after droughts ends, which captures aspects of both resistance and resilience. We agree that there was some confusion between the terms used along the manuscript and now clarified it following Vilonen et al (2022) recommendations: we are measuring the response of C and N fluxes during a post-drought period, and the term legacy effect is used to describe response during the post-drought period. We modified the terms along the manuscript accordingly and added a sentence in the introduction lines 122-124: *“Furthermore, while we have a good understanding of the response of ecosystem processes during drought, comparatively little is known about ecosystem responses, including C and N flux, after these events during a post-drought period³¹⁾”*

L69: I am not sure we are too curious as to how the relative abundance of fungi and bacteria modulate transfers of C from plants to belowground pools. Why the relative abundance?

There is several evidence in the literature that microbial communities in soil may shift from fungal to bacterial dominated one with intensive agriculture, so the statement of relative abundance was to refer to the commonly used bacteria/fungi ratio as an indicator of the soil food web properties (Bardgett & McAlister 1999, de Vries et al 2006, de Vries et al 2007). *Bardgett RD, McAlister E (1999) The measurement of soil fungal:bacterial biomass ratios as an indicator of ecosystem self-regulation in temperate meadow grasslands. Biol Fertl Soils 29: 282–290.*

De Vries FT, Hoffland E, van Eekeren N, Brussaard L, Bloem J (2006) Fungal/ bacterial ratios in grasslands with contrasting nitrogen management. Soil Biol Biochem 38: 2092–2103.

De Vries FT, Bloem J, van Eekeren N, Brussaard L, Hoffland E (2007) Fungal biomass in pastures increases with age and reduced N input. Soil Biol Biochem 39: 1620–1630.

We now clarify this in the introduction, see line 84: *“Such changes also have important consequences for biogeochemical cycles given that management-induced shifts in soil food web properties, including changes in the relative abundance of bacteria and fungi^{6,7,9}, can predict processes of carbon (C) and nitrogen (N) cycling^{10–12}. Specifically, shifts to bacterial dominated food webs resulting from management intensification have been linked to faster rates of nutrient mineralisation and greater losses of C and N from soil following perturbations such as drought^{10,13–15}”*

L70: belowground or below ground

This has been corrected as below-ground (and above-ground) throughout the manuscript

L108: intensification also meant more cattle per ha-1. So what about soil compaction?

Yes, intensive management may induce more compaction, which is why we measured soil bulk density as a proxy of soil compaction. Bulk density data were incorporated in the PCA presented in supplementary Fig. 6, and is now added in the supplementary Table 6 as well. Our primary aim was to consider the collective effect of grassland management on C and N fluxes in response to drought by measuring a range of soil properties relevant to these processes. Furthermore, manipulation experiments would be needed to disentangle all the proximate and ultimate effects of management on soil properties, but this was beyond the scope of this study. As noted above, a paragraph to acknowledge this has been added Lines 323-332.

L116: Please provide an international classification e.g. based on the World Reference Base for Soil Resources

This corresponds to Cambisol and has been added in the text (line 465)

Fig1: the figure suggests there are no trophic interactions between Microorganisms. What about bacterivore bacteria? What about nematodes and acanthamoeba feeding on bacteria and fungi? Thus I believe that the depiction of the expected flow of stable isotopes through the food web is a bit simple.

We now realise from the reviewer's comment that we did not explain the figure sufficiently well. To address this point, we now clarify caption for Figure 1 as follows: *“Conceptual diagram of the expected flow of ^{13}C (blue) and ^{15}N (yellow) through plants and soil trophic groups. The figure represents the groups we analysed for their ^{13}C and ^{15}N content, hence the simplification of the food web. The ellipses represent the trophic groups, and although interactions within trophic groups can happen, the main flow of stable isotope will be from one trophic group to another (represented by arrows). The isotopic enrichment is expected to decrease at higher trophic levels of the food web”*

1141: It does not become clear how many samples and replicates it was. Also where in the plots were these taken?

We clarify this in the text (line 476): *“The sample design is then 3 sites * 2 management type (extensive/intensive) * 2 treatments (control/drought) * 3 pseudo-replicates for a total of $n = 36$ plots”. A picture of the subplot and where the samples were taken has been added, see supplementary Fig. 10”*

L152: At what depth were these samples taken? How did the authors make sure that the sample received label in the end? Were those rhizosphere samples?

All the samples were taken from the surface to 7cm depth, and this is specified now in the method section (see line 510). Soil samples were bulk samples that included, but were not exclusively, rhizosphere soil that was sieved and homogenised. In pulse labelling experiments, it is very difficult to optimise timings of samples but we based our labelling timing and harvest time course on past experiments. This is now clarified in the text line 511: *“(based on time course from previous studies^{23,39,70})”*

FigS1: why did the soil moisture decrease more strongly in the extensively managed plots as compared to the intensively managed? Also it seems that the resistance of soil moisture to drought is higher in intensively treated plots.

From the Fig S1 (now supplementary Fig. 3) it seems indeed that soil moisture decreases more strongly in the extensively managed grassland. But this figure shows the soil moisture of two paired control and drought plots in one site only (extensive and intensive fields), so analysis of variance was not possible. A new supplementary figure has been added showing the average soil moisture content of all the plots across the 3 sites at 4 time points (See supplementary Fig. 4). From this data there is no significant differences of soil moisture reduction by the shelters between managements, but there is indeed a quicker recovery (i.e. trend towards control condition) of the soil moisture in extensively managed grassland. This is now clarified in the discussion section, see lines 288-293:

“Although drought tended to have a greater effect on soil moisture in the extensively managed grassland (trend of higher soil moisture reduction and significant slower recovery to the soil moisture observed in the controls), the effect of drought on C transfer was stronger in the intensively managed grassland. This finding indicates that the direction of the change in C transfer was not biased by the magnitude of the effect of the drought on soil moisture.”

L161: How were plant samples prepared for measurement of isotopic enrichment? Were samples homogenized? Same applies for soil samples

Samples (plant and soil) were homogenised and ground in a ball mill prior analysis; this has been added to the method section (line 524).

How much soil was taken overall (in grams) and how was it divided for the methods? It seems that 3 x 1cm cores do not yield a lot of soil material to begin with, how sure are the authors that the stable isotopes retrieved in those samples are representative of the soil food web? Also since the biomass of mesofauna was calculated

Only the initial sampling point at the end of the ^{13}C -CO₂ pulse labelling was done by pooling three 1cm x 7cm depth cores and was collected for PLFA analysis only. The subsequent sampling points, including collection of the mesofauna, were approximately 1/5th of the 40cm diameter collar up to 7cm depth, which corresponds to an average 200g of soil for the fauna extraction, with the rest sieved for chemical analysis. This was sufficient to collect soil fauna individuals to analyse their ^{13}C and ^{15}N contents (which needed 20ug of carbon minimum, so for example 50 individuals of collembola minimum). This information has now been clarified in the method section, see line 520-523, and the supplementary Fig. 2 shows the portion of soil sampled for each time point.

L284f: Please provide the values for the different parameters in a supplementary table. L116 says that soil had a similar pH of 5.5 but here you say that the pH was higher in intensively managed soils

The apparent inconsistency was due to the level of aggregation of the data: the pH stated in the methods section is an averaged pH measured from all plots to describe the main type of soil from our field sites. From the PCA in supplementary Fig.6, where the plot level values from control plots were used, pH is in general higher in intensively managed grassland

compared to extensively managed grassland. Field level values are now added in the supplementary table 1 as well and confirm that for 2 sites this is the case.

Fig2: Please provide a more extensive caption of what it shown. It does not become clear what “ento_bg” and other labels could stand for. Also, what are the PLFA data in there? Are they aggregated into groups? Also are the treatments drought and non-drought included as well as the sampling days?

The full caption was presented initially in the supplementary table 2 only as it took a lot of space but it has now been added to the original figure 2 (now Figure 3) as well. The PLFA data are from the control plots, which is specified in the caption, and the aggregation into main microbial groups (bacteria, fungi, AM fungi and actinobacteria) was specified in the method section.

L.301: I guess that ind.m-2 should stand for individuals per square meter but please introduce the abbreviation beforehand once. Also it does not make sense to have 3 digits for individuals? Also it does not become clear if these data come from the control plots and are averaged over replicates as shown in FigS4.

Ind. has been changed to individuals through the manuscript for clarity. A period will be used to indicate a decimal point, but here the comma is needed to separate every three digits (30,949 is thirty thousand nine hundred and forty-nine). These numbers correspond to an overall average from all plots and are here as an indicator of the community structure in general in our systems. This has been now clarified in the text lines 197-200.

L301-303: “Collembola was the most abundant group...” But of what?

This has been specified: “*Collembola was the most abundant group of mesofauna*” (line 198).

L307: PERMDISP

This change has been done

L317: “¹³C pulse labelling of plant shoots”. It occurs to me that the atmosphere was labeled where the plant would fix the labeled CO₂. Thus, no labeling of the plant shoots was performed. Is this correct?

This is correct and ‘of plant shoot’ has been removed

L334: “Plant leaves in...” There is no mention of plant leaves in the figures and the methods so far. Were leaves measured instead of shoot material? Again describe in the methods

Plant shoots were collected and analysed as specified in the method. “Plant leaves” has been replaced by “plant shoots”.

L344: True but the label found in these groups were also much lower than in other groups. Especially for AMF and bacteria, the incorporation of label is very low. Why could this have been the case?

This was already discussed in the discussion section, see line 353-358:

“The greater ^{13}C -enrichment of non-mycorrhizal fungal PLFA compared to the AM fungal PLFA is initially surprising, and implies an important role of non-mycorrhizal fungi in channelling plant-derived C into the soil food web, supporting other recent findings^{38–41}, and reflecting that saprotrophic fungi form a significant portion (20–66 %) of microbial biomass in a grassland rhizosphere⁴². However, it may also reflect that ^{13}C is turning over rapidly in AM fungi⁴³, and being lost as respiration^{44,45}..”

Fig. 6 seems to be missing

This was to refer to the Figure 7. We apologise for the misnumbering in the figures, which has now been corrected (it now refer to Figure 6, the schematic of the main results).

L. 410: The authors found an effect of drought on grassland management types, and this effect differed with regard to plant C assimilation and allocation belowground. However, how can the authors be sure that these effects are not simply due to the plant biodiversity and/or the soil moisture?

We are unable to unequivocally separate proximate and ultimate effects of management on C flow and its resistance to drought in our study, and this was not the aim of this experiment. However, we can say that intensive management, including all the changes that this management implies in terms of soil compaction, soil moisture, plant biodiversity, soil biodiversity etc., does reduce the C assimilation and C allocation to certain component of the food web and its resistance to drought in general. A statement to clarify this has been added lines 323-332: *“The classification of grassland in this study into extensive and intensive management regimes captures a suite of above- and below-ground properties. Although we measured differences in key properties between the grassland types (Fig 3; supplementary Figs. 4 and 6), we are unable to unequivocally separate proximate and ultimate effects of management intensification on below-ground C flow. There are many factors involved in grassland management, including fertiliser use and differences in grazing intensity and compaction associated with livestock and machinery use³³. All these factors have potential to impact soil abiotic and biotic properties and vegetation (Fig 3; supplementary Figs. 4 and 6) and disentangling their impact on below-ground C fluxes in response to drought would require additional manipulation experiments in the field, which was beyond the scope of this study.”*

L456: I do not see how this statement is justified by the data presented. There is not causal relationship between soil activity and photosynthetically derived C studied here. The data does not allow the authors to judge on the total activity of participants within the system from stable isotope tracing as performed here.

We understand the reviewer’s comment and agree that we cannot demonstrate that causal link, although we think it is appropriate to propose what the causal link could or could not be given the data at hand, and offer an interpretation of the data. A relatively higher proportion of ^{13}C was found in the soil CO_2 efflux in the extensively managed grassland compared to the intensively managed grassland. Although we agree that we cannot draw conclusions on the total activity of participants within the system, soil respiration is often used as an indicator of the ‘global’ soil activity. Therefore, a higher proportion of ^{13}C in soil CO_2 efflux indicates that in comparison of the overall activity, a higher proportion of ^{13}C -recent photosynthates derived carbon is processed below-ground. We modified the sentence to explain better our statement, see line 383-387:

“Our results show that soil CO₂ efflux is relatively less enriched in ¹³C in the intensively managed compared to extensively managed grassland in control conditions. This finding indicates that in intensively managed grassland, soil activity (including roots and soil organisms) relies proportionally less on C derived from recent photosynthate compared to extensively managed grasslands.”

Reviewer #2 (Remarks to the Author):

This study addresses the interaction between drought and management intensity in grasslands, monitoring ¹⁵N and ¹³C stable isotope signature through several levels of the soil food web over time. It is a very ambitious experiment that involves measurements in three paired sites over 20 days and a large array of scientific expertise. It is a very solid piece of work.

I suggest clarifying the way knowledge gaps are presented, since for the moment they are described in sequence with little integration: 1) how shifts in food web structure and f:b ratio modulate C transfer to the soil and plant nutrient capture(L69); 2) need for a multitrophic perspective to address the response of C and N losses to perturbation (L76); 3) response of rhizodeposition to changes in land use (L82). They also suffer from syntax issues, especially when listing items (L83 soil C and N cycling and sequestration; L78 the responses of processes of C and N loss to perturbations, such as drought; L79 awareness of the importance of rhizodeposition as a driver of... and the structure and functions of...; L89 incorporation of ... into... and CO₂ and N₂O fluxes...).

Improving the structure is not only important to clarify the gaps, but also to specify where the study fills them, and get back to that in the discussion.

The f:b ratio deserves some sort of introduction, as it appears out of nowhere. Further, I am not sure that “shifts in bacterial and fungal energy channels” (L66) is clear for all readers of this journal, please explicit or rephrase.

We thank the reviewer for their comments and thoughtful suggestions. The introduction has been modified to improve the structure concerning the knowledge gaps and hypothesis. The f:b ratio is now better introduced in the introduction (lines 92-95), and we have removed the term “channels” and now use “bacteria- or fungi-dominated food web” (lines 88-89).

I assume that the authors chose to leave it out of the scope of the study, but from a topological point of view, I am left wondering how soil food web structure and complexity is affected. Are the nodes and connectance modified, is there any rewiring? I think it is relevant to address this structural point, because it should be related to the fluxes of C and N that are the focus of the study.

While we are aware of other studies linking the structure of soil network to faster nutrient cycling (Morriën et al, 2016), this is beyond the scope of this study. Our objective was to measure C and N fluxes to key components of the food web, which correspond to aggregates of species within Collembola, detritivorous mites, other detritivores, predatory mites, other predators, bacteria and fungi. These soil community groups are too broad to reliably and meaningfully quantify network parameters such as connectance. We added a paragraph to the discussion section and suggest that future studies should look the food web structure at a finer level, see lines 316-322:

“We suggest that modification of the food web structure and composition at finer resolutions, or an enhanced efficiency of nutrient transfer within the soil food web, could drive feedbacks between plant and ecosystem functioning and their response to drought. Further work is needed to understand how shifts in the composition of complex food webs formed in the natural environment, such as those observed in the current study, feedback to plants over longer time periods and regulate ecosystem processes.”

Interaction between management intensity and drought is the heart of this study in my view. The results start with a focus on management and drought, the interaction with drought appearing in part 4. I think it would be nicer to explicitly state in the results that the interaction was significant before diving into the different responses under different management regimes, to keep a focus on the interaction rather than on the management treatment. This suggested structure is actually followed in the discussion, so restructuring the results would fit better with it.

We have reordered the result section as suggested, putting the interaction between management intensity and drought first.

I enjoyed the discussion, it is concise and to the point. Several aspects could still be improved. 1) It tries to explain everything at times, which is expected given the impressive dataset, but could be reduced. For example, L459 less C input belowground was not measured yet presented as such. 2) It sometimes opens arguments that are not resolved: L448 what timeline are we considering for the C retention? L427-4299 what is the authors' position considering the apparent contradiction?

The discussion section has been modified according to the reviewers' comments and as a result we think it is much improved. Concerning the first point raised, C input below-ground has been measured and we observed lower C recovery in roots, AMF and mites in intensively compared to extensively managed grassland (see supplementary Fig. 9). Concerning the second point, the arguments mentioned by the reviewer are now better discussed, see lines 370-372: *“Our results confirm that fungi have a more important role in extensively managed than intensively managed grassland¹⁴ and that they promote the retention of recently assimilated C in soil^{28,38}, at least during the time scale of this study..”* and 343-350: *“Despite large differences in plant biomass allocation, management intensity had no detectable effect on root ¹³C enrichment, indicating a similar rate of root C allocation of newly incorporated photosynthates in both management regimes. This contrasts with the general idea that slow-growing plants allocate more C to their roots³⁵, and potentially indicates a rapid transfer of root-¹³C to below-ground organisms, in line with plants with thick roots diverting C to collaboration with soil organisms^{36,37}.”*

The conclusion goes for generalisation, yet it is a difficult position to hold regarding CO₂ and N₂O emissions as they were measured in this study. N₂O emissions are typically very transient, and would have to be measured right after the rain events that occurred at the end of the experiment to catch a bump. It looks like it was the case on day 10, was it significant? More generally, how robust is the generalization of N₂O and CO₂ emissions response based on half a dozen point-in-time measurements over 20 days, especially when relating to generalisation of N₂O production L472?

We agree with the reviewer's comment, but our aim was not to provide a quantification of emissions from the different managements, in which case we would have measured more intensively as the reviewer eludes to. Rather we measured fluxes as indicative of fate of applied ^{13}C and ^{15}N resulting from changes in transfer pathways under different management and drought. To recognise these points, we have added caveats to the discussion and conclusion section to discuss the robustness of the N_2O emissions. The additions are as follows: (in the discussion section lines 406-410) *“The production of N_2O in soils is typically heterogeneous leading to hot-spots and hot-moments⁶¹, and in extensive grassland there was marginally greater emission of N_2O (which we attribute primarily to the nitrate reducing processes of denitrification and nitrate ammonification) immediately after the ammonium nitrate was injected into the soil and during the following day (supplementary Fig. 7).”*; and in the conclusion (lines 433-435)

“The transient nature of N_2O fluxes and the variety of mechanisms contributing to its production prevent us from providing robust conclusions on its response to drought and management.”

Given the large dataset, could the authors identify in their study, and propose, a marker variable for specific ecosystem services? This could simplify the endeavour they propose in the last sentence of the conclusion.

Many studies have sought to identify indicators for healthy soils, and there is a consensus about the need to include biological and soil biodiversity indicators (Lehmann et al. 2020). Although we agree that such markers are desirable in the context of grassland management and drought, our work did not set out to do this. Nevertheless, we highlight that consideration of aspects of C movement from the plant to belowground organisms could have potential as indicators of how grasslands perform under different perturbations. This is now discussed in the conclusion section, see line 441

“Although the movement of C belowground cannot be used as a specific indicator of soil health⁶⁷, it may be a good indicator of how systems perform in response to perturbations.”

Other comments

L35 photosynthates

This change has been done

L41 “However” contrasts N transformations against C transfer, which is a bit awkward in itself, and this sentence about N is also lost in the middle of a section in C transfer.

This sentence has been removed

L43 disrupted, L45 provided

These changes have been done

L35, 46, 47 photosynthates

These changes have been done

L45 “impacts to drought” please rephrase

This sentence has been removed

L53 Shifts in soil food web structure both regulate the stability of soil functions and impair their resistance. This is awkward wording.

This sentence has been removed

L55, 45 I don't understand why "ability to buffer" is used instead of "resist to" as in stability theory.

This sentence has been removed

L59 "These drivers" please explicit

This has now been specified 'A major challenge is to understand the interactions between management intensification and drought'

L66 "including..." does not fit in the sentence

This sentence has been rephrased, see line 84-87:

"Such changes also have important consequences for biogeochemical cycles given that management-induced shifts in soil food web properties, including changes in the relative abundance of bacteria and fungi^{6,7,9}, can predict processes of carbon (C) and nitrogen (N) cycling¹⁰⁻¹²."

L150 change to consecutive

This change has been done

L363 change in function to as a function

This change has been done

L373 change its to ¹³C

This change has been done

L462 change to through

This change has been done

L468 change to marginally

This change has been done

L452-454 this information is general and its location does not participate in driving the discussion. Please restructure the entire paragraph L449-454.

We are not entirely sure what the reviewer's point is here, but we are open to more revisions if the reviewer could give us some more indications. We think this paragraph is important as it develops a hypothesis to explain the fact that drought only affected ¹³C transfer to certain

trophic groups (bacteria and collembolans only), and have done some rewording to clarify this as follows (lines 373-381): “Drought decreased the flow of ^{13}C in the intensively managed grassland through several components of the food web, in particular bacteria and Collembola. This could be explained by the fact that Collembola are more sensitive to drought than other soil fauna and often reduce their activity in response to drought²³ and bacteria grow quickly and are more sensitive to drought and other stresses than fungi⁴⁶⁻⁵¹. Furthermore, some fungi are able to create large hyphal networks that facilitate nutrient and water transfer over long distances, and indirectly benefit plants by exploring water-filled soil pores not accessible to plant roots^{50,52,53}. Although unlikely, it is also possible that species producing extensive networks could have been influenced by conditions outside the experimental plots.”

Figures: please explicit variable names throughout the manuscript and supplemental, correct superscript/subscript for isotopes/gases and scientific notation of units. Please include units in all figures, even supplemental (S1, S3).

These changes have been done

Fig. 7 is actually Fig. 6

This change has been done

Fig. S3 requires units and change efflux to efflux rate

These changes have been done

Fig. S5 explicit legend

This has been modified and is more explicit now

Fig. S7 change stock to pool size, along time to over time. Explicit that the numbers at the top are days after labelling.

These changes have been done

Reviewer #3 (Remarks to the Author):

The authors describe an impressive experiment in which they manipulate drought in two types of grassland communities – extensively and intensively managed – and quantify carbon and nutrient transfers throughout the entire food web. Their results suggest that intensive grassland management has a negative influence on the diversity of the belowground food web (lower dispersion) and the resistance of carbon transfers belowground following drought, which they link to this disrupted food web. Drought led to reduced aboveground biomass in intensively managed grasslands but not extensive grasslands and the isotopically labeled carbon was concentrated aboveground with reduced transfers belowground in the intensive grassland. The paper is the first of its kind that I have read and was well written. However, there are some issues that I have with the methods and interpretation of results that need to be resolved prior to publication.

First, the authors should emphasize that they are measuring grassland community resilience or recovery from drought. No measurements were made during the drought treatments but rather in the days following re-watering.

We thank the reviewer for their supportive comments and suggestions for further improvements, which we have addressed, as detailed below. As suggested we added a stronger emphasis on the definition of what we are measuring throughout the manuscript following Vilonen *et al.* (2022) recommendations, i.e. the response of C and N fluxes during a post-drought period, the term legacy effect is used to describe response during the post-drought period, see lines 122-127:

“Furthermore, while we have a good understanding of the response of ecosystem processes during drought, comparatively little is known about ecosystem responses, including C and N flux, after these events during a post-drought period³¹”

Vilonen et al. (2022). What happens after drought ends: synthesizing terms and definitions. New phytologist 235(2):420-431. doi: 10.1111/nph.18137

Second, the 20-year management practices have led to significant changes to above and belowground community composition. Obviously management has caused these differences, but is there a way to disentangle the specific effects of community change on the C and N flows you describe?

Although it would be indeed very interesting, our study does not allow us to disentangle the proximate or ultimate effects of management on C and N flow and its resistance to drought, and this was not the aim of this experiment. As discussed in response to reviewer 1, further studies would be needed to separate the effect of each proximate and ultimate effects of land management like soil compaction, soil moisture, plant biodiversity, soil biodiversity etc. A paragraph has been added in the discussion to clarify this point, lines 323-332.

Third, the introduction could use a bit more detail on how isotopically labeled C and N have previously been used to answer similar questions.

Finally, the interpretation of C13 results is puzzling to me as plant C13 is often a metric of water use efficiency. C13 naturally occurs in the air and leaves/shoots with a high concentration of C13 generally reflect high water use efficiency as stomata close and discrimination against the heavier isotope decreases. Do the differences you observe between these two grassland types in terms of C13 accumulation simply reflect differences in water stress? For example, greater C13 accumulation was observed in the intensive grassland during drought which experienced greater biomass reductions during drought. I'm unfamiliar with the methods used for concentrating gas around the plant tissue, but my assumption is that this would also influence leaf physiology simply by increasing ppm of CO₂ around leaves which leads to greater photosynthesis and eventual stomatal closure (leading to increased C13 accumulation). Is the concentration of C13 in the gas samples high enough that discrimination can be ignored and all carbon taken up is C13? Are some of these issues resolved by comparing the C13 in labeled to unlabeled controls? Either way, some discussion of how stomatal functioning influences vegetation C13 accumulation is warranted.

Concerning the discrimination of ¹³C and the difference of the stomata closure between species, we agree that this can happen and the difference in natural abundance $\delta^{13}\text{C}$ of leaves can be observed between species (Padilla et al 2019). However, the scale of the change in $\delta^{13}\text{C}$ is minimal compared to the amount of ¹³C we introduced to the system. As an

example, Padilla et al 2019 found that severe drought reduced significantly the ^{13}C signature of the plant leaf by a maximum of 2 ‰ after 3 years of drought. By comparison, we increased the $\delta^{13}\text{C}$ signature of the plant shoot after ^{13}C pulse labelling from $-30 \pm 0.1\text{‰}$ to $462 \pm 34\text{‰}$. The natural content of ^{13}C in unlabelled controls (from each field) has indeed been subtracted from the ^{13}C content of each sample to remove the potential difference of ^{13}C signature between management and sites. This was made clear in the method section, line 595:

“The isotopic concentration data was converted from $\delta^{13}\text{C}$ and $\delta^{15}\text{N}$ values (‰) to atom % excess ^{13}C and ^{15}N by subtracting the atom % ^{13}C and atom % ^{15}N of unlabelled controls from each enriched sample (Johnson et al., 2011)”

*Padilla et al. (2019). Effects of extreme rainfall events are independent of plant species richness in an experimental grassland community, *Oecologia* 191:177-190.*

Specific comments:

Line 41: change to lowered the rate of reduction

This change has been done

Line 111: Extensive fields were not cut for hay?

No, they were not, this has been added to the text

Line 117: Here would be a good place to mention sample design. From my reading, you have $n=6$ plots (3 control, 3 drought) in each field per site, correct? So, a total $n=9$ for the following treatments: “drought intensive”, “drought extensive”, “control intensive”, “control extensive”. But really, your level of replication is site, so $n=3$ for each?

The sample design has been summarised, see lines 476-478

Line 117: Also, note that these roofs were 100% exclusion. Most drought shelters are passive reductions, making this unique in that respect. See Knapp et al. (2017). Pushing precipitation to the extremes in distributed experiments: recommendations for simulating wet and dry years. *Global Change Biology*, 23(5), 1774-1782.

Most of the studies (72%) reviewed by Knapp et al (2017) used a passive approach to reduce rainfall inputs based on a relative reduction in ambient precipitation amounts. However, many of these studies were partial passive reduction all year round to simulate extreme precipitation years (reduction shelters), while our shelters were passive rain exclusion by 100% only in the summer to simulate extreme weather events (exclusion shelters), which are predicted to increase in frequency, intensity and duration by climatic models. These two approaches are used to answer different questions and are complementary. The type of exclusion shelters and our method of implementation have been frequently used by others (e.g. de Vries et al. 2012 *Nature Climate Change*; de Vries et al. 2018 *Nature Communications*; Fry et al, 2018 *Ecology*; Fuchslueger et al, 2016 *Journal of Ecology*; Ingrisch et al, 2019 *Global change biology* etc.). This is now clarified in the text lines 466-475:

“Most of the studies (72%) reviewed by Knapp et al. used a passive approach to reduce rainfall inputs based on a relative reduction in ambient precipitation amounts⁶⁸. However, many of

*these studies were partial passive reduction all year round to simulate extreme precipitation years (reduction shelters), while our shelters reduced rainfall by 100% only in the summer to simulate extreme weather events (exclusion shelters), which are predicted to increase in frequency, intensity and duration by climatic models. In each field, an extreme drought event was simulated by placing passive rain exclusion shelters which reduced rainfall by 100%⁶⁸. Three transparent roofs (1.5 m * 1.3 m) alongside delimited control plots were set-up for 60 days (17-19 of May – 17-19 of July 2016) designed to simulate a 100-year drought event⁶⁹.”*

Line 137: Change to “5 hours after watering the plots”

This change has been done

Line 140: You injected the annual total N in one go? Is that appropriate? Also, this is significantly lower than the amount of N that the intensive grassland usually received correct?

Yes, the reviewer is correct in that the N has been added in one event, following the approach used in many other ¹⁵N labelling studies (Guardia et al 2018; Murphy et al, 2017; Mörrien et al, 2016). Although we agree that this can cause artefacts associated with large nutrient additions, we kept the amount added as small as possible to minimise these effects, while being sufficient to trace the ¹⁵N into soil organisms. As a comparison, the intensively managed grassland typically received more than 100kg ha⁻¹ year⁻¹ while we injected the equivalent of 20 kg ha⁻¹, which is 5 times lower to what the intensively managed grassland receives annually, and which correspond to what the extensively managed grassland receives in a year by annual N deposition. We added some caveat in the text line 494-496: *“The quantity of N injected (equivalent to just 20 kg N ha-1.) was kept as small as possible to minimise potential artefacts associated with large nutrient additions, whilst still enabling quantification of the tracer in soil organisms.”*

Line 144-145: Could you include an image of this set up?

A new figure has been added to show the set-up, see supplementary Fig.2

Line 148: Does this amount of labeled carbon increase ppm of CO₂ surrounding the plant? This can lead to increased photosynthesis followed by stomatal closure in some plants if so. Additionally, does this treatment completely replace natural C¹²-CO₂ in the chamber with labeled C¹³-CO₂? Or do the plants still discriminate against the heavier isotope.

We followed well established methods for pulse labelling with ¹³C which was applied at 99 atom % ¹³C-CO₂ (Johnson et al 2005, Chomel et al 2019, Pang et al 2021). We agree that elevated CO₂ concentrations in the chamber may have a small impact on plant physiology, but our approach using multiple releases of ¹³C likely minimises this effect. Prior to labelling, measurements of the photosynthetic rate were done for this reason to determine the timing of injection to limit excessive or deficient CO₂ concentrations in the chamber. In terms of discrimination between isotopes, this is trivial compared to the relative proportion of ¹³C in the chamber.

Chomel et al. 2019. Drought decreases incorporation of recent plant photosynthate into soil food webs regardless of their trophic complexity. Global Change Biology 25, 3549–3561.

Johnson et al 2005. Soil invertebrates disrupt carbon flow through fungal networks, Science vol 309, issue 5737 p1047.

Pang et al 2021. In-situ ^{13}C labelling to trace carbon fluxes in plant-soil-microorganism systems: Review and methodological guideline. Rhizosphere20 (2021) 100441

Figure 2: Are only the significant variables shown in the PCoA? Also, I recommend making the labels clearer or including the meaning of the abbreviations in the figure legend rather than the supplementary material.

Yes, this was specified in the legend *'and variables with $P < 0.05$ were kept'*

We added the abbreviations meaning in the legend of the Figure 2 (now Figure 3)

Line 301: Consider re-wording the start of this paragraph. Needs a topic sentence or lead-in to what "group" you're talking about. "In terms of soil community composition,..."

This has been modified as suggested

Line 283: Yes, at least for control plots.

We think this comment refers to line 383. The statistical results of the mixed model are: Treatment $P = 0.05$, Management $P = 0.04$ and interaction Mgmt * Trt $P = 0.07$ meaning there is no significant interaction between management and treatment. For this reason, we cannot say that this is true only for control plots.

Line 284-286: This is not clear to me from Figure S7. Can you show a summary stat across treatments? A more detailed description of what high C13 or N15 implies would also be nice.

We think this comment refers to line 384-386. A new supplementary table with the stats summary corresponding to supplementary Figure S7 (now supplementary Fig. 10) has been added (supplementary Table 5). The reviewer can see that grassland management decreases detritivorous mite ^{13}C and ^{15}N pool size, predatory mites ^{13}C and predatory fauna ^{13}C pool size. These stats are mirroring the stats in sup Table 4 (for the supplementary Fig 9), which is why we initially choose to not integrate them, but we agree that if the figure is presented, we need to present the statistics as well.

Line 306: This dispersion result is particularly interesting as a metric of diversity and should be discussed more in the discussion.

We thank the reviewer for his suggestion to develop more this result. However, at the level of definition of our soil communities (mainly families and trophic groups) we cannot expand on a metric of diversity per se. However, this result can indicate that intensive management determines homogenisation of the relative abundance of the main soil fauna groups. We added few sentence in the discussion, see lines 392-397:

"Although our results showed only small differences in the food web structure, at least at the level of definition we studied, we found a higher dispersion of the soil communities in the extensively compared to intensively managed grassland. This result could indicate that intensive management determines homogenisation of the relative abundance of the main soil fauna groups, potentially due to homogenisation of soil properties in space⁵⁵."

Line 369-370: Does the lack of C13 enrichment reflect continued drought stress and the

plants not opening stomata to take in C13? Stomatal closure varies dramatically between species

There is no lack of C13 enrichment, but there is a lack of difference between droughted or control in the extensively managed grassland, while in the intensively managed grassland drought increased or decreased the C13 enrichment of different pools compared to the control plots. We added ‘compared to the controls’ for clarity, see line 176.

Line 411: No drought impact on aboveground biomass in extensive grassland is notable and should be mentioned here. Perhaps the drought was not severe enough

We applied the same drought perturbation in both management types based on previous experiments in the same region (De vries et al, 2018; De Long et al 2019), which is quite severe for this region of the UK as it represents a 110-year drought event. As we can see a similar effect of the drought shelters on soil moisture reduction, or even a trend for stronger reduction of soil moisture in extensively managed grassland, we are confident that this perturbation was severe enough and allows us to compare these two systems. We modified the sentence for clarity lines 287-299: *“The legacy effect of drought on plant C assimilation and its allocation below-ground differed between the two grassland management types. Although drought tended to have a greater effect on soil moisture in the extensively managed grassland (trend of higher soil moisture reduction and significant slower recovery to the soil moisture observed in the controls), the effect of drought on C transfer was stronger in the intensively managed grassland. This finding indicates that the direction of the change in C transfer was not biased by the magnitude of the effect of the drought on soil moisture. During the post-drought period in intensively managed grassland, and despite decreases in above-ground biomass, plant shoot C uptake increased and C transfer to roots, bacteria and collembolans decreased compared to controls. However, in extensively managed grassland, drought had no significant legacy effect on the plant biomass or C transfer below-ground (except to collembolans).”*.

Line 453: This is a good point – should be specific here about how these biota could access areas outside of the drought shelters as the soil was not hydrologically isolated from the surround soil matrix.

We added a sentence line 379 to be more specific: *“Although unlikely, it is also possible that species producing extensive networks could have been influenced by conditions outside the experimental plots.”*

Furthermore this point was already discussed earlier in the discussion section (see lines 303-306): *“Our finding is consistent with recent reports of greater resistance and faster recovery of plants in abandoned grassland relative to managed grasslands due to larger below-ground root and fungal networks in the former, which improves water access compared to intensively managed grasslands”*

Line 487: Doesn’t this go against your recent findings (Chomel et al. 2019) that drought acts independently of trophic interactions to influence carbon storage aboveground and flows belowground? Perhaps discuss the differences between these two papers in greater detail.

This is now discussed in the discussion section Line 307-322:

“However, our recent work, which manipulated components of grassland food webs under controlled laboratory conditions, did not find any interactive effects of drought and food web

composition on below-ground C flow and plant growth suggesting feedback effects may be of limited importance²³. The modest duration, unique plant species and limited pool of soil faunal species used in that study may have precluded the development of tight interactions among plants, soils and soil organisms that limited feedback effects. In the present study, grassland management influenced below-ground C flow and there were relatively small differences in food web composition, at least at the resolution measured (i.e. no differences in the biomass of microorganisms and only differences in biomass of some decomposers between the management regimes; Fig. 3 and supplementary Fig. 8). We suggest that modification of the food web structure and composition at finer resolutions, or an enhanced efficiency of nutrient transfer within the soil food web, could drive feedbacks between plant and ecosystem functioning and their response to drought. Further work is needed to understand how shifts in the composition of complex food webs formed in the natural environment, such as those observed in the current study, feedback to plants over longer time periods and regulate ecosystem processes.”

Figure 7. This is a very helpful figure but is never cited within the text. Notably, drought had a positive influence on C13 storage in tissues. How do you tease apart radiolabeled carbon 13 that you added from natural carbon 13 that is known to accumulate in drought stressed plants (particularly c3 plants) as stomata close and discrimination of heavier isotopes is reduced? Is this determined from the difference between non-labeled control tissue? Also, in figure 7 it is difficult to determine whether an arrow in the food web is referring to C or N transfers, yet these were different in their responses to drought.

There was a misnumbering, and this figure is now Figure 6, which is cited Line 282. Concerning the discrimination of ¹³C and stomata closure, we agree that this can happen in nature, however the scale of this change is minimal compared to the enrichment of ¹³C we add to the system. As an example, Padilla et al 2019 found that severe drought reduced significantly the 13C signature of the plant leaf by a maximum of 2 ‰ after 3 years of drought. By comparison, we increased the δ13C signature of the plant shoot after 13C pulse labelling from $-30 \pm 0.1\text{‰}$ to $462 \pm 34\text{‰}$. We modified the Figure 6 legend to clarify the arrows significance.

Figure S1: There seem to be clear differences in the effectiveness of the shelters between extensive and intensive. Wasn't the point of watering to remove the effect of drought? In which case, the plots should have similar soil moisture after shelters are removed. At what depth was soil moisture measured?

From the reviewer's comment we understand that this was unclear, and we now explained it better in the manuscript. Our aim was to study C and N flow post drought, in the idea to capture aspects of both resistance and resilience. The re-wetting approach ensured plots contained photosynthetically active plants that would capture 13C, and thus avoid the risk of missing C and N transfers in the droughted treatment, in addition to capture a critical phase of the community response to drought (see Vilonen et al, 2022). This is now made clearer in the introduction, see lines 122-124:

“Furthermore, while we have a good understanding of the response of ecosystem processes during drought, comparatively little is known about ecosystem responses, including C and N flux, after these events during a post-drought period³¹”

The effectiveness of the shelters, and the recovery of soil moisture after drought, is independent of the local climatic conditions and variation in soil type, as the paired sites were next to each other. This means that the difference in C and N transfers could be due to the ultimate effects of the land management on these systems, i.e., plant species composition, organic matter content, soil compaction etc. Controlling the effectiveness of shelters *in situ* between plots and sites is extremely challenging and would require a more complicated set-up. Instead we wanted to study how grassland management, including all the changes that it causes, shape the response of C and N cycles to a similar drought perturbation. A statement to clarify this has been added lines 323-332:

“The classification of grassland in this study into extensive and intensive management regimes captures a suite of above- and below-ground properties. Although we measured differences in key properties between the grassland types (Fig 3; supplementary Figs. 4 and 6), we are unable to unequivocally separate proximate and ultimate effects of management intensification on below-ground C flow. There are many factors involved in grassland management, including fertiliser use and differences in grazing intensity and compaction associated with livestock and machinery use³³. All these factors have potential to impact soil abiotic and biotic properties and vegetation (Fig 3; supplementary Figs. 4 and 6) and disentangling their impact on below-ground C fluxes in response to drought would require additional manipulation experiments in the field, which was beyond the scope of this study.”

Note that the data shown in supplementary Figure 3 is from moisture probes installed on one pair of plots in one site only, as we did not have the capacity to install datalogger in all plots and all sites. However, the trend is similar in other plots, and we added data from sequential measurements made at several time points at all the sites; see supplementary fig. 4. Statistical analyses on these data show no difference in moisture reduction in both management but slower recovery of the soil moisture in extensively managed grassland compared to intensively managed grassland.

Continuous moisture measurements were made with the SM300 moisture probe from Delta-t, having two pins of 51mm that have been inserted in the soil. Soil moisture measurements at each sampling event were measured with the WET-2 sensor probes from Delta-t with pins of 68mm inserted in the soil. These details have been added in supplementary methods.

Vilonen et al 2022. What happens after drought ends: synthesizing terms and definitions, New phytologist.

Reviewer #4 (Remarks to the Author):

The authors present an interesting field study on the impacts of land management on soil food web responses to drought, with a focus on fluxes of C and N among plants, microbes and microfauna. In this study, extensively and intensively managed grassland sites were first exposed to a long period of summer drought, after which C and N fluxes were studied using isotope tracing. Most importantly, the authors showed that after drought, carbon transfer to the belowground system was limited in intensively managed systems but not in extensively managed systems. The study thereby shows another clearly negative implication of agricultural intensification on ecosystem functioning. I agree with the authors that this is an important finding. However, I do have a number of comments regarding the data analysis and presentation of the results, as well as some other suggestions to further improve the manuscript.

General comments

1) My first comment relates to Figure 2 and the corresponding PERMANOVA analysis. It is unclear to me which samples exactly are represented by the individual dots in the figure, as based on the experimental design, I would expect single soil community composition measurements per (sub)plot (18 per management treatment/36 in total). However, there are more samples shown in the PCA plot. To avoid pseudoreplication, I would suggest to only incorporate a single sample per plot (as done in Fig. S2), at least when the goal is to just examine the effect of management on community composition. However, I think it would also be insightful to distinguish between drought and non-drought plots in these analyses, as drought will likely have caused a shift in community composition, the strength of which may have depended on management.

These data correspond to the data from control plots only to study the initial community structure and its modification by land management, but at several sampling points. So $n=18$ plots * 5 time points = 90. This has now been specified in the legend.

The data in supplementary Figure 2 (now supp. Fig. 6) was measured only at the initial sampling point, hence the difference of numbers of points.

2) The authors used t-tests to analyse whether drought positively or negatively affected nutrient transfers to different response groups. However, with these analyses differences in drought effects between intensively and extensively managed fields are not explicitly tested, while in my opinion these comparisons are also important. Moreover, by using t-tests it's not possible to incorporate among-site variation, which may have obscured treatment effects. Therefore, I would suggest to analyse the log response ratios using mixed models as well. To understand whether the response ratios are significantly different from zero, confidence intervals of predicted means can be examined.

We thank the reviewer for the suggestion, and have done the new statistical analyses as suggested: we applied a mixed model with management as fixed factor and plot within site as random factor and extracted the 95% confidence intervals of predicted means. The new statistical analysis is detailed in the method section and the Figure 2 and supplementary table 4 have been modified accordingly. Two main differences appeared: the logRR from Collembola ^{13}C enrichment is different from 0 in the extensively managed grassland and the logRR from actinobacteria ^{13}C enrichment is different from 0 in the intensively managed grassland. These differences do not change the main conclusion, as drought has stronger effects on several ^{13}C transfers to different C pools in intensively managed grassland (plant shoot, plant roots, bacteria, actinomycete, collembola and CO_2 efflux) while it only has effects on Collembola ^{13}C enrichment and ^{15}N enrichment of N_2O efflux in extensively managed grassland.

3) While I understand that the used methodology limits the number of sites that can be examined, I think it would be good to acknowledge that the study only included a single soil type, and that studies in different soil systems are needed to test the generality of these results.

This has now been acknowledged in the conclusion, see line 443: *“This highlights the need for future studies on a range of soil types to examine trade-offs of grassland management intensities for various climate change scenarios...”*

4) To improve accessibility to a wider audience, I would recommend the authors to expand the description of the examined food web. I think Figure 1 shows this food web well, but in the text the different food web components are not explicitly mentioned. Furthermore, key components of soil food webs, most notably protists and nematodes, have not been analysed, I assume due to methodological limitations. I think this is worth some lines of discussion.

In parallel of the Figure 1, the different food web components are detailed in the method section:

“The samples were further grouped into 7 main trophic groups in order to have sufficient material to analyse ¹³C and ¹⁵N: detritivorous Collembola, detritivorous mites (oribatid, astigmata and prostigmata mites), annelids, other detritivorous (detritivorous coleoptera, myriapoda and diptera larvae), herbivores (hemiptera and thysanoptera) predaceous mites (mesostigmata and predaceous prostigmata) and predaceous fauna (arachnida, chilopoda, predatory coleoptera and symphyla).”

Specific comments

L59: Please specify ‘these drivers’ in order to further clarify the goal of the study in this sentence.

This change has been done

L63-64: AMF also are soil biota, so perhaps write: ‘...decrease the abundance and diversity of soil biota, including arbuscular mycorrhizal fungi, and induce shifts in soil microbial community composition.’

Given that your study has a multi-trophic perspective and includes more than just microbes, it would be good to expand the second part of the sentence by highlighting that also non-microbial groups in the soil community are affected (e.g. micro-arthropods).

These changes have been done and the sentence expanded:

“...decrease plant diversity, the abundance and diversity of larger body sized earthworms, nematodes and microarthropods, or the biomass of saprotrophic and arbuscular mycorrhizal (AM) fungi.”

L64-68: I suppose ‘including shifts in the relative abundance of bacterial and fungal energy channels’ refers to ‘soil food webs associated with agricultural intensification’, but this is not completely clear. Do you mean ‘changes in soil food webs associated with agricultural intensification’?

This paragraph has been rewritten, see lines 94-91:

“Such changes also have important consequences for biogeochemical cycles given that management-induced shifts in soil food web properties, including changes in the relative abundance of bacteria and fungi^{6,7,9}, can predict processes of carbon (C) and nitrogen (N) cycling^{10–12}. Specifically, shifts to bacterial dominated food webs resulting from management intensification have been linked to faster rates of nutrient mineralisation and greater losses of C and N from soil following perturbations such as drought^{10,13–15}”

L106: 'and of similar topography' refers to the 'three geographically distinct sites', right? In that case I would suggest to write 'three geographically distinct sites of similar topography were selected'.

Similar topography referred to the paired fields, but this sentence has been rewritten to improve clarity: "*Three geographically distinct sites were selected, each with adjacent, paired grasslands (field) on the same soils and of similar topography but with different long term (>20 years) history of intensive or extensive grassland management (see supplementary Fig. 9 and supplementary Table 6 for more details).*" (lines 452-455).

L270-272: If I understand it correctly, the drought treatment was not included as a fixed effect in these models? I think it would be insightful to do this, even when drought effects are separately presented in the analyses of response ratios.

The reviewer is right in the fact that drought treatment was not included as a fixed effect in these models because only the data from control plots were used. This choice has been made to avoid redundancy with the analyses of response ratios, and because we wanted to study the effect of land management on the C and N fluxes, without taking into account the potential interactive effect of drought.

L273-280: It would be good if the calculations done to produce Fig.S5 are also explained here. Furthermore, I assume that adjacent control and drought plots were used to calculate independent replicate logRRs, but it would be good to specify this.

This has now been specified line 636-639:

"...calculated as the log Response Ratio from each paired plots (logRR), which quantifies the proportional difference between mean ¹³C-, ¹⁵N-enrichment or biomass in control and drought conditions"

L301: 'most abundant group of mesofauna'

done

Caption Fig. S5. It's unclear on which calculation the presented metrics are based. I thought they would be purely based on measured (PLFA-based) biomass, but I got confused by L70 ('on the ¹³C transfer'). Could you please clarify this here and in the methods?

That was a mistake from copying the caption of Figure 5 and has been now corrected in the legend

Reviewer comments, second round review -

Reviewer #1 (Remarks to the Author):

I thank the authors for this thorough revision which clarified many of my previous concerns. I was very critical in the first round of review and I am happy to see that the authors were able to address almost all of the comments. It is an extensive study that will enhance our knowledge of drought legacy effects on an ecosystem level. Moreover, as the authors state in the revised version, it is hardly possible to disentangle the individual effects/players in such a complex system.

Reviewer #2 (Remarks to the Author):

I find the new version of the manuscript improved in clarity, with all my comments and questions addressed. This is a very nice piece of work.

Reviewer #3 (Remarks to the Author):

The authors have addressed my comments made on the previous version of the manuscript. It is an impressive dataset and the authors have done a good job interpreting the nuances of these complex systems.

I have no further comments on the paper.

Reviewer #4 (Remarks to the Author):

I thank the authors for carefully addressing the reviewer comments. The manuscript has been strongly improved in terms of clarity and structure, making it a very interesting read for a broad readership.

I only have a few minor editorial suggestions:

L87: 'play'

L88: It is currently not completely clear what 'their ability' refers to.

L307: I think 'during nature restoration' is superfluous here, given that it's already mentioned in the first part of the sentence.

L315-316: Perhaps specify that this is especially is to be expected for mycorrhizal fungi?

Authors response to Reviewer Comments -

Response to reviewer 4: L87: 'play' Response: This change has been done L88: It is currently not completely clear what 'their ability' refers to. Response: we change the sentence with "...the ability of soil food webs..." L307: I think 'during nature restoration' is superfluous here, given that it's already mentioned in the first part of the sentence. Response: this has been removed L315-316: Perhaps specify that this is especially is to be expected for mycorrhizal fungi? Response: this has been added: "Furthermore, some fungi, in particular mycorrhizal fungi, are able to create large hyphal networks"